

# Accurately assessing congenital heart disease using artificial intelligence

Khalil Khan[1], Farhan Ullah[2], Ikram Syed[3] and Hashim Ali[1]

[1] Department of Computer Science, School of Engineering and Digital Sciences, Nazarbayev University, Astana, Kazakhstan
[2] College of Computer Science and Software Engineering, Shenzhen University, Shenzhen, China
[3] Dept of Information and Communication Engineering, Hankuk University of Foreign Studies, Yongin, Gyeonggy-do, Republic of South Korea

## ABSTRACT

Congenital heart disease (CHD) remains a significant global health challenge, particularly contributing to newborn mortality, with the highest rates observed in middle- and low-income countries due to limited healthcare resources. Machine learning (ML) presents a promising solution by developing predictive models that more accurately assess the risk of mortality associated with CHD. These ML-based models can help healthcare professionals identify high-risk infants and ensure timely and appropriate care. In addition, ML algorithms excel at detecting and analyzing complex patterns that can be overlooked by human clinicians, thereby enhancing diagnostic accuracy. Despite notable advancements, ongoing research continues to explore the full potential of ML in the identification of CHD. The proposed article provides a comprehensive analysis of the ML methods for the diagnosis of CHD in the last eight years. The study also describes different data sets available for CHD research, discussing their characteristics, collection methods, and relevance to ML applications. In addition, the article also evaluates the strengths and weaknesses of existing algorithms, offering a critical review of their performance and limitations. Finally, the article proposes several promising directions for future research, with the aim of further improving the efficacy of ML in the diagnosis and treatment of CHD.

## INTRODUCTION

### Background

Congenital heart disease (CHD) represents a significant global health challenge, impacting approximately 1% of live births around the world (*Boneva et al., 2001*; *Rosamond et al., 2007*; *Pierpont et al., 2007*). CHD encompasses a variety of structural anomalies in the heart and blood vessels, which develop before birth. These defects can vary widely in severity, from simple problems that can be resolved alone to complex malformations that require extensive medical intervention. Some common forms of CHD include pulmonary atresia, critical aortic stenosis, and hypoplastic left heart syndrome, among others.

Corresponding author
Khalil Khan, khalil.khan@nu.edu.kz

The implications of CHD are profound, as the condition can lead to a spectrum of long-term health problems, including arrhythmias, developmental delays, and even mortality, if not diagnosed and managed effectively. Early detection and accurate diagnosis are crucial to improve outcomes. Traditional diagnostic methods, such as fetal echocardiography (EKG) and parental ultrasound (US) (*Chew et al., 2007*; *Murphy et al., 1975*; *Friedberg et al., 2009*; *Ma et al., 2023*; *Jiang et al., 2023*; *Reddy, Van den Eynde & Kutty, 2022*; *Li et al., 2019*), although effective, are often limited by accessibility, especially in resource-poor settings. The challenges in early and accurate diagnosis require innovative solutions to improve the prognosis and quality of life of affected individuals.

CHD refers to a range of birth defects that affect the structure and function of the heart. These anomalies arise due to the inappropriate development of the heart during fetal growth. CHD can manifest in various forms, each differing in severity and impact on the individual's health. The primary types of CHD include:

*Pulmonary atresia (PA):* Characterized by an obstruction or complete closure of the pulmonary valve, impeding blood flow from the right ventricle to the lungs. Severe forms are associated with an underdeveloped right side of the heart, known as hypoplastic right heart syndrome (HRHS).

*Critical aortic stenosis (CAS):* Involves significant narrowing of the aortic valve, which restricts blood flow from the left ventricle to the aorta and the rest of the body. This condition often accompanies hypoplastic left heart syndrome (HLHS), where the left side of the heart is underdeveloped.

*Hypoplastic left heart syndrome (HLHS):* A severe condition where the left side of the heart is critically underdeveloped. This defect severely limits the heart's ability to pump blood efficiently, often leading to inadequate mixing of oxygenated and deoxygenated blood.

These congenital defects can lead to significant morbidity and mortality, particularly in newborns, necessitating early diagnosis and intervention. The visual presentation of normal heart and CHD is given in Fig. 1. In particular, fetal CHD has recently emerged as the leading cause of mortality among all congenital disabilities. This increase in fetal CHD mortality has made it the leading cause of infant mortality, placing increasing burdens on families and societies alike. In China, infants diagnosed with CHD comprise up to 6%–8% of all live newborn births. According to statistical reports, an estimated 150,000 infants with CHD are born annually in China, almost 30% experiencing complications and exhibiting a poor prognosis, making them currently untreatable with satisfactory results. Therefore, early detection and screening for fetal CHD are of paramount importance.

## Role of AI in CHD

Artificial intelligence (AI), particularly through the use of machine learning (ML) and deep learning (DL) techniques, has shown tremendous potential to transform the landscape of CHD diagnosis and treatment. AI systems can analyse vast amounts of complex medical data, including imaging, genetic, and clinical records, to identify patterns and make predictions that may be beyond human capabilities. This capability is particularly valuable

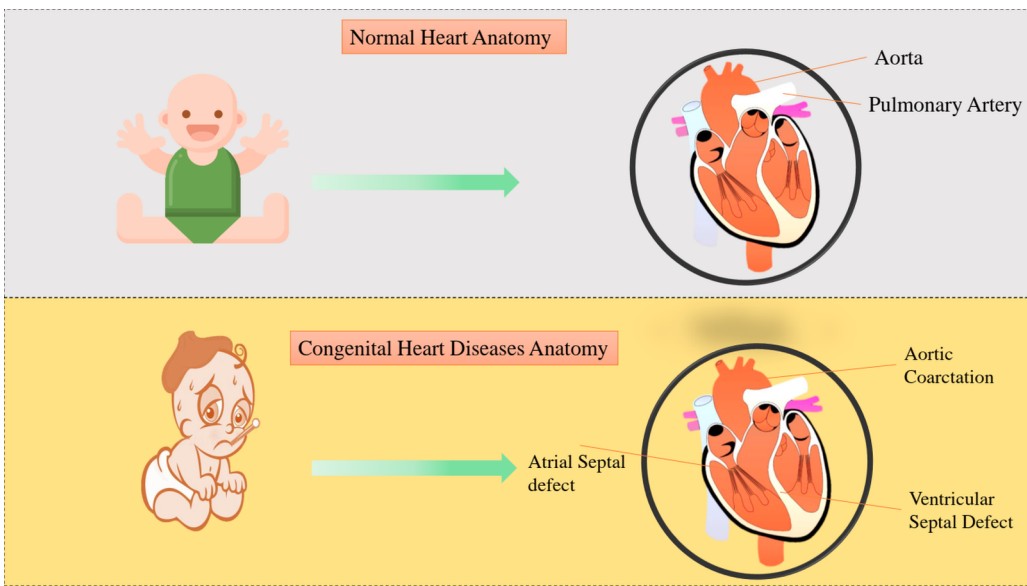

**Figure 1** Normal cardiac anatomy in children is shown and is contrasted with the anatomical variations observed in CHD.

in the context of CHD, where subtle anomalies in heart structure and function must be detected with high precision.

The application of AI in CHD research has evolved rapidly over the past decade. Early AI models focused on basic pattern recognition and statistical analysis, but recent advances have introduced sophisticated neural networks capable of DL from data. These AI models are now being used to improve imaging techniques, predict disease progression, and assist in surgical planning and interventions. For instance, convolutional neural networks (CNNs) have been used successfully to look at echocardiograms and find structural heart defects. Recurrent neural networks (RNNs), on the other hand, are used to understand time series data, like ECGs.

Despite these advances, the integration of AI into clinical practice for CHD is still in its infancy. Challenges related to data quality, interpretability of AI models, and regulatory hurdles must be addressed to fully realize the potential of AI to improve CHD outcomes.

## Objectives of the survey

The primary objective of this survey is to provide a comprehensive review of the application of AI in the diagnosis and treatment of CHD in the past 8 years. This survey aims to cover a broad range of AI techniques used in CHD research, including ML, DL, and hybrid models. Similarly, this article has focus on renowned research articles that have made significant contributions to the field. Studies were selected based on their relevance, impact, and novelty in applying AI to CHD.

The article is structured to first present the background and significance of CHD, followed by an overview of AI techniques employed in CHD research. Subsequent sections

will discuss the applications of these techniques in various aspects of CHD treatment, identify key challenges and limitations, and explore future directions for research.

By synthesising the findings of a decade of research, this survey aims to highlight the progress made, identify ongoing challenges, and propose future research directions to improve the role of AI in combating CHD. The ultimate goal is to provide a valuable resource for researchers, clinicians, and policy makers to understand the current state of AI in CHD and to promote further advancements in this critical area of healthcare.

## SURVEY METHODOLOGY

This article presents a summary of CHD using ML in general and DL in particular. This work is the first to provide a thorough overview of the CHD using ML techniques, based on our current understanding. The research article focuses on the work that has been done for CHD recognition using ML in the last eight years (2016–2024). As we know, there has been a shift from traditional ML to DL in recent years. Therefore, our work will also try to focus more on the methods that are using DL. Furthermore, a comprehensive and structured discussion will be provided for each topic, adhering to a systematic approach. This is outlined in Fig. 2, which visually represents the organisation of our discussion.

The primary contributions offered by the proposed article are as follows: Firstly, the article discusses all the sources of signals that physicians and medical practitioners use to analyze CHD in newborn babies. Similarly, the article presents the importance of each source, highlighting its contribution to the recognition of CHD. Secondly, the article uses some standard databases (DBs) to evaluate any ML/DL model. All the DBs that are used for CHD recognition are reported in the literature and discussed in this article. Third, the application of AI for CHD recognition presents several significant challenges. The article reports all these problems in detail. We also discuss some strategies for solving these problems. Fourth, even though ML-based CHD recognition is not a well-explored area, researchers have made good progress in the last couple of years. This article discusses and reports on each of these methods. We also discuss the pros and cons of each method. Lastly, the discussion concludes with reports and methods for CHD recognition using ML/DL. We also discuss potential avenues for researchers to explore this field in future work.

We followed a systematic selection of studies for this review article. We conducted an extensive search on academic DBs such as IEEE Xplore, ACM Digital Library, ScienceDirect, and Google Scholar. We use keywords such as "congential heart disease", "machine learning", "deep learning", "artificial intelligence", "Cardiac Imaging", "Echocardiography", and "Diagnosis and Prognosis of CHD". We restricted our searches to articles published between 2016 and 2024. This help us to capture the most recent developments in the field. We also performed forward and backward citation analysis to identify further relevant articles that we may missed in the initial search results. Initially, more than 450 results were obtained from the search. These results were then evaluated for relevancy by comparing the titles and abstracts with the main subject of this survey. Excluding duplications, a final set of extremely pertinent articles were chosen for examination. Furthermore, the reference lists of these articles were thoroughly examined to

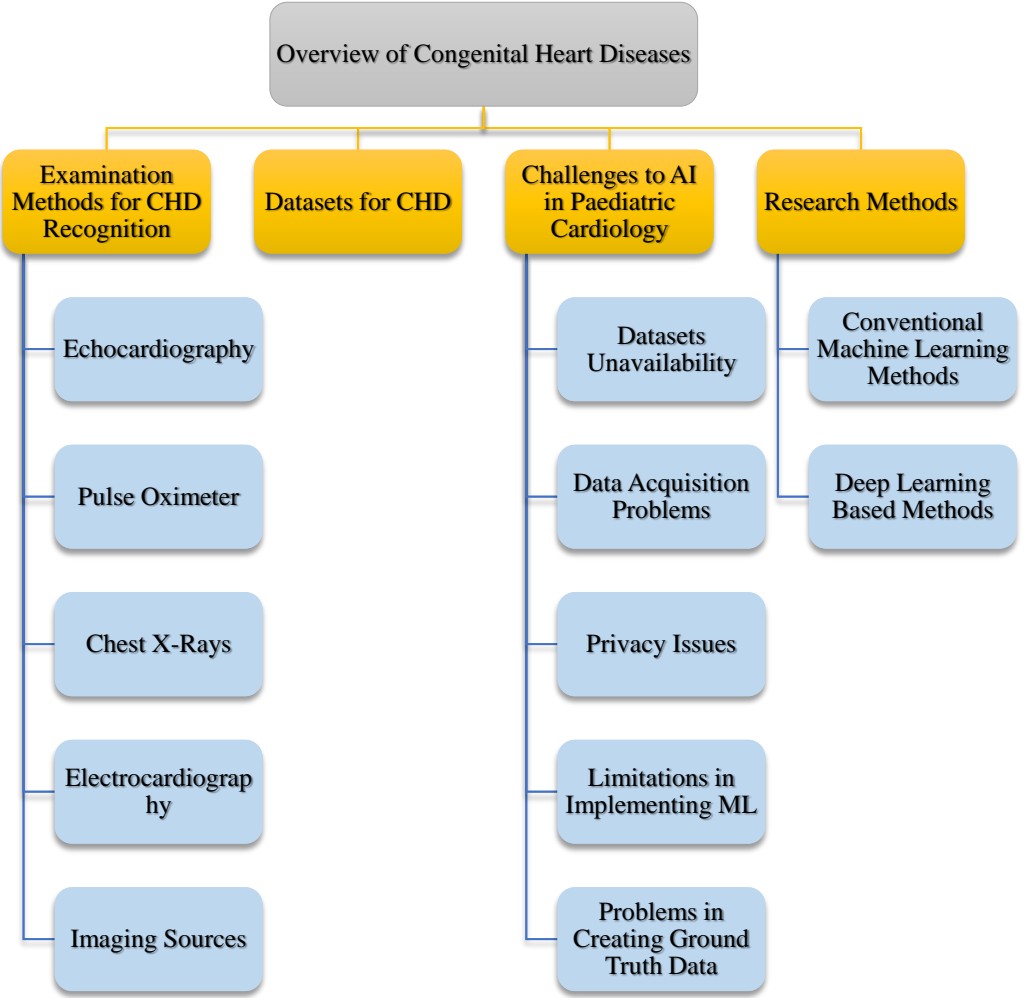

**Figure 2** A comprehensive overview of congenital heart disease is shown, with a systematic discussion of each topic involved, presented in a step-by-step manner.

detect any supplementary sources that were pertinent to the topic. The chosen publications were carefully examined to extract essential information regarding the methods. Ultimately, the limits and areas of research that need further exploration were combined to provide a well-rounded viewpoint.

In the propsed article, we applied rigorous exclusion and inclusion criteria. We included studies if the articles met certian criteria. Some of the inclusion and exclusion rules were as follow:

- We focused on articles where ML/DL was the central methodology.
- The proposed article explicitly addressed the application of ML and DL algorithms to CHD recognition. We did not consider heart problems for aged people. We excluded studies that focus on adult cardiovascular diseases.

- We collected information about all validated DBs reported in the literature. None of the DBs reported so far are synthetic in the literature.
- While including articles, we also selected some quantitative measures such as accuracy, sensitivity, specificity, F1-score, or AUC.
- We included only peer-reviewed articles in our study. We excluded preprints not subjected to peer review. We excluded review articles, editorials, and commentaries, which did not present original research.
- We excluded all studies published in languages other than English.

## CURRENT DIAGNOSTIC METHODS

This section discusses the various diagnostic techniques used by medical professionals to analyse CHD. CHD is a common form of birth abnormalities and a significant contributor to children's illness and death (*Zimmerman, 2020*; *Roth et al., 2018*). Rapid and precise identification of afflicted paediatric patients is critical for prompt treatment and successful surgical outcomes (*Tworetzky et al., 2001*; *Bonnet et al., 1999*; *Van Velzen, 2015*; *Morris, 2014*). Diagnostic methods such as transthoracic echocardiography (TTE), X-rays, cardiac magnetic resonance imaging (MRI), and dual-source CT exams are very popular, but have complicated steps, take a long time, cost a lot and need to be performed by experienced cardiologists (*Corbett, 2022*). Unfortunately, there is a widespread occurrence of delayed diagnosis, even in instances that require urgent attention. This leads to a less ideal clinical response, particularly in places with lower incomes. A study carried out in a low-income country has shown that the delay rate can reach 85%.

Due to excellent ML models and the establishment of well-defined datasets for CHD, the entry of AI-based methods into pediatric diagnostics is now possible. In this section, we examine data sources that may help identify CHD. Medical professionals use most of these sources, either individually or in combination for CHD recognition.

### Fetal ultrasound

This is the most commonly used prenatal screening method, allowing visualization of the structure and function of the fetal heart. Despite its widespread use, fetal ultrasound (US) has limitations in resolution and may miss certain defects.

The most widely used diagnostic technique for prenatal screening is the fetal US method, which allows visualization of the structure and function of the fetal heart. Despite significant advances in fetal US imaging technology, the prenatal detection rate for fetal CHD remains notably inadequate, as evidenced by population-based clinic studies, primarily due to several challenges (*Pierpont et al., 2007*; *Chew et al., 2007*). Firstly, fetal four-chamber (FC) views often exhibit lower resolution, increased speckles, and artifacts, which presents considerable obstacles for cardiologists in diagnosing CHD. In FC views, the physical boundaries between the four chambers may not be clear or may not be present at all, especially during the moments of opening the mitral valve, the tricuspid valve, and the atrium. In such cases, the resemblance among the four chambers becomes remarkably high, necessitating a heavy reliance on the cardiologist's experience with chamber identification. Lastly, variations in sonographer experience levels and the fetal position within the uterus

can introduce inconsistency and lack of repeatability in acquiring US images, further challenging cardiologists in fetal CHD diagnosis. Because of these big problems, a good system for diagnosing CHD needs to be able to quickly pick up features that don't change based on location or situation and are specific to each person.

## Fetal echocardiography

This is a specialized form of ultrasound that provides detailed images of the fetal heart. Although highly effective, it is labour-intensive and requires specialized expertise, limiting its availability and use. According to medical practitioners and reported research, this is the most authentic signal for assessing CHD in a patient's heart (*Oh, 2007*; *Shamsham & Mitchell, 2000*; *Kirkpatrick et al., 2007*). However, the main issue with echocardiography (EKG) machines is that they are not available everywhere. Furthermore, the proper diagnosis of EKG signals requires expert knowledge. These experts are also not readily available anywhere, especially in rural areas. Although an EKG is considered the most reliable method for diagnosing CHD, it typically requires around 10 min to complete the diagnosis. The new AI-based technologies have almost revolutionized modern medicine when it comes to obtaining an EKG with great precision. Combining AI-based methods with EKG signals can help create an efficient prediction system that does not require extensive knowledge of the fetal EKG. Some methods that use this combination are listed in *Oliveira et al. (2021)*, *Oliveira et al. (2018)*.

## Pulse oximetry

It is a non-invasive method that measures oxygen saturation levels in the blood. It is often used in conjunction with other screening techniques to identify potential CHD cases. Although an EKG is the most reliable method for diagnosing CHD, it typically requires more time and expert knowledge with the equipment for analysis. Alternatively, POX is user-friendly and can assess findings in a short time frame of 2 to 3 min. Figure 3 shows how Pulse oximetry (POX) can be used to see the output of a pulse. Since its introduction and establishment, the CHD screening method *via* POX has been progressively implemented in clinical practice (*Q-m et al., 2014*). In addition, the use of POX as a supplement to current standard practice is expected to be a cost-effective approach based on generally accepted benchmarks (*Roberts et al., 2012*; *Thangaratinam et al., 2012*).

Newborns typically undergo POX screening within 24 to 48 h after birth as a non-invasive and cost-effective technique to detect serious CHD. Despite its usefulness in identifying serious CHD in neonates, this screening method cannot consistently identify all forms of CHD. POX screening may overlook certain types of congenital cardiac disease because they may not cause substantial alterations in oxygen saturation. The EKG, which is widely considered the most reliable technique for detecting CHD, often requires approximately 10 min to complete and may not be practical for every infant, especially in areas with limited resources. On the other hand, POX offers a convenient and efficient option, as it only takes 2-3 min to examine the findings (*Suvorov et al., 2023*).

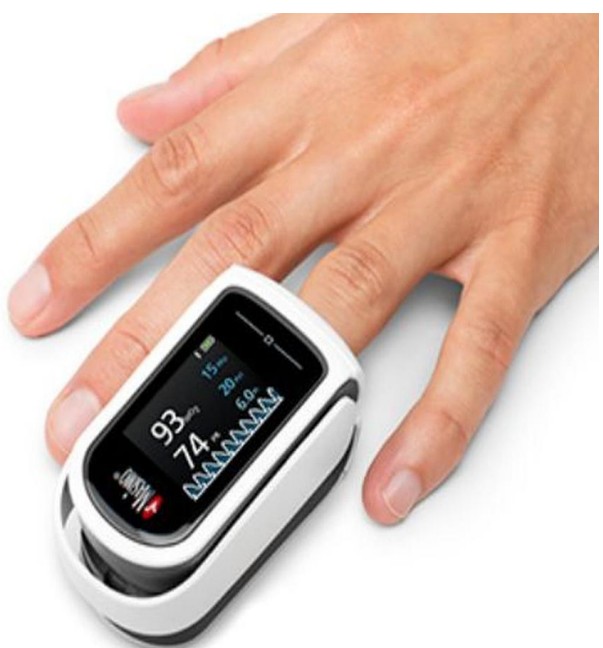

**Figure 3** The operational mechanism of a pulse oximeter.

## Chest X-rays

It is used to detect structural abnormalities in the heart and lungs. Chest radiographs can provide initial information, but are often insufficient for a complete diagnosis of CHD. There are some classic X-ray signs that can help with CHD recognition (a comparison can be seen in Fig. 4. Some of the signs include a boot-shaped heart, a Smith sign, an egg on a string, a gooseneck sign, a figure of three (coarctation of the aorta), and a box-shaped heart. The good thing about chest radiographs is their easy acquisition process. It is simple and readily available, allowing for a direct diagnostic approach to CHD recognition. We believe that chest radiographs are not very helpful in CHD recognition; however, initial scanning with a chest radiograph is the easiest method. Once we diagnose certain abnormalities, we can refer patients for further investigation.

## Electrocardiography

The electrocardiography (ECG) records the electrical activity of the heart and aids in the detection of arrhythmias and other electrical abnormalities. Although valuable, ECG interpretation requires significant expertise. ECG has become a widely adopted tool for prenatal diagnosis, due to its inherent safety, cost effectiveness, non-invasiveness, and ability to provide real-time imaging (*Tworetzky et al., 2001*). In particular, ECG stands out for its effectiveness in assessing both the structure and function of the fetal heart, playing a crucial role in both the diagnostic process and the ongoing treatment of fetal CHD; Ultrasound imaging stands as an indispensable tool. Within the realm of prenatal imaging, the four-chamber (FC) view of the fetal heart is of exceptional importance. This specific perspective serves as the cornerstone of the diagnosis of fetal CHD and is widely used by

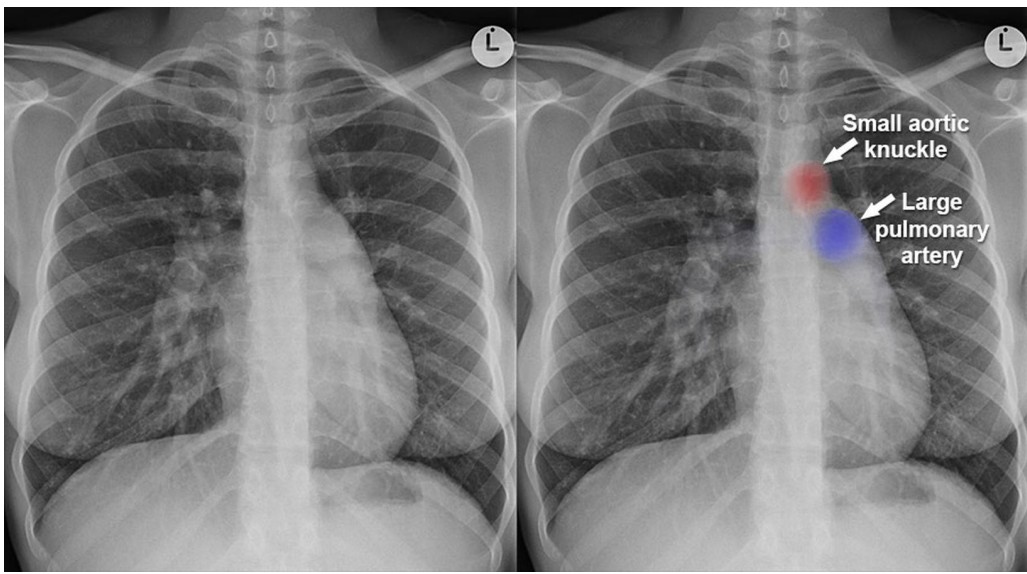

**Figure 4** **Comparison of chest X-ray images of a healthy child and a child with CHD.** For more information, visit https://www.radiologymasterclass.co.uk/gallery/chest/cardiac_disease/congenital_heart_disease.

clinicians during prenatal evaluations. Using this primary ultrasound image, clinicians gain unparalleled insight into the intricate developmental morphology of the fetal heart. This direct visualisation allows meticulous examination of cardiac structures and dynamics, enabling clinicians to detect anomalies early and devise customised treatment strategies.

Analysing fetal FC views to diagnose fetal CHD requires a profound understanding of the fetal cardiac anatomical structures among cardiologists. However, accurate identification of fetal heart disease is a knowledge-intensive endeavour due to the intricate nature of the structures of the fetal heart, leading to a long learning curve. Consequently, training cardiologists proficiently in the diagnosis of fetal CHD can be both costly and time consuming.

Adult cardiology is in some sense, a well-studied and well-explored area by computer vision and ML researchers. Some relevant articles can be studied in *Dong et al. (2019a)*, *Acharya et al. (2019)*, *Olanrewaju et al. (2021)*, *Li et al. (2020)*, *Jahmunah et al. (2024)*, *Cheng et al. (2024)*. Heart failure is a critical condition characterised by the inability of the heart to adequately provide the body with enough oxygen and nutrients for optimal functioning. Rapid identification and precise diagnosis of heart failure are crucial in preventing the progression of the condition.

An ECG is a diagnostic procedure that captures the electrical activity and rhythm of the heart, mainly used for the detection of heart failure (example, Fig. 5). The ECG detects abnormalities in heart rhythm or electrical conduction and assesses the presence of previous heart attacks, ischemia, and other diseases that can trigger heart failure. However, there are instances where it becomes challenging and time-consuming to decipher the ECG signal, even for a proficient cardiac specialist. ECG signals are also used for heart diagnosis in

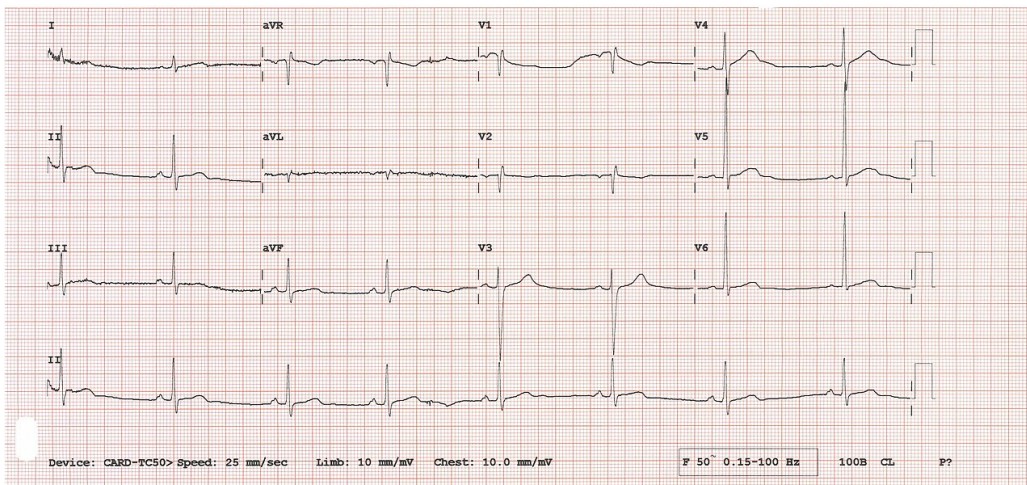

**Figure 5** **The ECG of a heart affected by CHD, followed by a detailed interpretation of the ECG output results.** The analysis will highlight key abnormalities and diagnostic indicators specific to CHD.

infants. Some of the methods that use ECG signals are discussed in detail in 'Research Gaps and Future Directions'. ML and DL-based methods are used in conjunction with ECG signals. We argue that the ECG is a stable signal that can give enough information about the heart abnormality. However, screening for every one is not possible with ECG. Furthermore, there are some major issues with the acquisition of the ECG signal that are not covered in this article.

## Imaging sources

Alternative imaging modalities, such as computed tomography, computed tomography angiography, and cardiac magnetic resonance imaging, can be used to identify CHD in newborns. However, these methods are not the main imaging modalities for this purpose. Magnetic resonance images of cardiac patients have also been reported as methods that are often used for structural and functional evaluation of the heart (*Yuan et al., 2021*; *Kholmatova, Kharkova & Grjibovski, 2016*; *Ng et al., 2022*). The MRI acquisition technology has grown over the years from some conventional methods such as cardiac gating to the latest and most advanced methods of high-field strength magnets and ultra-fast pulse sequences.

In the field of MRI, the advancement and novelty of AI has allowed very short scan times. By minimising motion artifacts due to the movement of patient, this also reduces errors. Segmentation of cardiac chambers enhances visualization and aids in diagnosis. Currently, medical professionals perform the majority of this segmentation process manually. If this is shifted to AI-based methods, diagnosis will be faster and this will also reduce variations between different practitioners (*Kholmatova, Kharkova & Grjibovski, 2016*).

## Challenges in current diagnostics

Despite advances in diagnostic technologies, several challenges persist in the detection and treatment of CHD:

*Resolution and accuracy:* Prenatal imaging techniques like fetal US and EKG often suffer from lower resolution and artifacts, making it difficult to obtain clear and accurate images of the fetal heart.

*Expertise requirement:* Techniques such as fetal echocardiography and ECG require specialized knowledge and experience, limiting their accessibility, especially in low-resource settings.

*Inconsistency and variability:* The accuracy of CHD diagnosis can vary based on the experience of the sonographer and the position of the fetus, leading to inconsistent results.

*Resource limitations:* In middle- and low-income countries, limited access to advanced diagnostic tools and specialized medical personnel exacerbates the challenge of early CHD detection and treatment.

*Delayed diagnosis:* Often, CHD is not detected until significant symptoms appear postnatally, which can delay critical interventions and adversely affect outcomes.

Addressing these challenges through innovative approaches, such as the integration of AI and ML into diagnostic processes, holds promise for improving early detection and treatment of CHD.

## AI TECHNIQUES FOR CHD DETECTION AND TREATMENT

AI, particularly through the use of DL and ML techniques, has revolutionized the healthcare sector, particularly in the detection and treatment of CHD (*Liu et al., 2022*; *Tan et al., 2023*). AI systems can analyze vast amounts of complex medical data, including imaging, genetic, and clinical records, to identify patterns and make predictions that may be beyond human capabilities. This capability is particularly valuable in the context of CHD, where subtle anomalies in heart structure and function must be detected with high precision. This section provides an overview of various AI techniques used for the detection and treatment of CHD, highlighting the advances and methodologies used.

The application of AI in CHD research has evolved rapidly over the past decade. Early AI models focused on basic pattern recognition and statistical analysis, but recent advances have introduced sophisticated neural networks capable of deep learning from data. Deep learning (*Bengio, Goodfellow & Courville, 2017*), a subset of machine learning, refers to a class of models based on artificial neural networks with many layers (often called deep neural networks) that are designed to automatically learn complex patterns from large datasets (*Ullah et al., 2024*). Unlike traditional machine learning methods that rely on handcrafted features, deep learning models can extract high-level representations directly from raw data, making them highly effective in tasks such as image recognition (*He et al., 2016*), natural language processing (*Deng, 2018*), and time-series analysis (*Chen et al., 2023*). These AI models are now being used to improve imaging techniques, predict disease progression, and assist in surgical planning and interventions.

Recent research by *Komatsu et al. (2021)* proposed SONO, an architecture that uses CNNs to discern cardiac abnormalities observed in fetal US videos. This method utilizes timeline visualization to assess the probability of detection and compute anomaly scores. The evaluation focuses primarily on cardiac structural anomalies, specifically heart and

vessel anomalies, employs AUC-ROC analysis, and shows better performance relative to established methodologies. Similarly, *Gangadhar et al. (2023)* explored the viability of leveraging DL techniques, particularly artificial neural networks, for the prediction of early stage coronary artery disease. The objective of the study is to improve cardiac diagnosis and precautionary actions by effectively analyzing data patterns. In addition, (*Gonsalves et al., 2019*) investigated the use of cardiac datasets to predict CHD through various ML methods. Their study highlights the capacity of naive Bayes probabilistic algorithms to improve the detection of CHD anomalies.

CNNs stand as a quintessential algorithm in DL. The use of CNNs in medical image processing is widespread, due to its ability to learn high-level features with enhanced distinguishability and robustness, thus effectively representing images. For example, *Kuruvilla & Gunavathi (2014)* introduced simple feedforward and backpropagation CNNs to classify lung cancer in CT datasets. Their approach achieved remarkable accuracy. Similarly, *Song et al. (2018)* unveiled a sophisticated multitask cascade architecture of CNNs (MTC-CNN) specifically tailored for the automated recognition and detection of thyroid anomalies. Their innovative approach marks a significant advancement in image analysis, promising a more efficient and accurate diagnosis of thyroid-related conditions. In particular, CNNs have become indispensable tools in automating the diagnosis of the prevailing COVID-19 pandemic, as mentioned in research work by *Wu et al., (2021)*, *Farooq & Hafeez (2020)*, *Zhou, Canu & Ruan (2020)*. This underscores the critical contribution of CNNs in addressing the urgent need for efficient and accurate diagnostic solutions amidst the global health crisis.

Furthermore, obtaining high-resolution fetal FC views holds paramount importance not only for accurately estimating fetal abdominal circumference but also for effectively diagnosing fetal deformities. This emphasizes the indispensable role of clear and detailed FC views in comprehensive prenatal evaluations and early detection of potential fetal abnormalities. Several previous studies have successfully obtained standard views of fetal FC with essential anatomical structures through a thorough analysis using CNN (*Wu et al., 2017*; *Bridge, Ioannou & Noble, 2017*; *Dong et al., 2019b*). Despite these advances, diagnosing fetal CHD poses significant challenges. To address this, previous research efforts have proposed diagnostic systems that aim to improve the diagnostic accuracy of CHD from 65% to 85% (*Rocha et al., 2013*; *Van Velzen et al., 2016*; *Gong et al., 2019*). However, these systems often encounter two notable drawbacks: first, they may exhibit relatively lower accuracy, which can undermine their credibility in fetal CHD diagnosis; second, their diagnostic processes often resemble a CNN's "black box", insufficiently clear explanations provided to cardiologists while supporting them in medical diagnosis, thereby hindering effective understanding and decision-making processes.

## DATASETS FOR CHD RESEARCH

The performance of any ML-based method depends on the dataset used for experimental validation and testing. In this part of the article, we present an overview of the datasets introduced so far for CHD recognition using ML. In recent years, the CHD recognition

**Table 1  CHD databases reported in SOA.**

| S. No. | Dataset | Year | Total applicants | Data types | Dataset available? |
|--------|---------|------|------------------|------------|--------------------|
| 1 | ZCHSound (*Liu et al., 2022*) | 2024 | 941 | PCG | Yes |
| 2 | CHDdECG (*Chen et al., 2024*) | 2024 | 65,869 | EKG | No |
| 3 | DICOM (*Gerke, Minssen & Cohen, 2020*) | 2024 | 828 | X-rays | Yes |
| 4 | PhysioNet (*Garcia-Canadilla et al., 2020*) | 2024 | 33 | ECG | Yes |
| 5 | CMUD (*Chen et al., 2024*) | 2022 | 475 | PCG | No |
| 6 | POX (*Roberts et al., 2012*) | 2022 | 44,147 | Saturation level | No |
| 7 | UCI (*Arooj et al., 2022*) | 2022 | 1,050 | EKG | Yes |
| 8 | HSS (*Qiao et al., 2022*) | 2019 | 170 | PCG | Yes |
| 9 | Doppler TTe (*Hrusca et al., 2016*) | 2018 | 1,932 | ECG | Yes |
| 10 | CXRAY (*Seah et al., 2019*) | 2016 | 46,712 | X-rays | No |
| 11 | HVSMR | 2016 | 12 | EKG | Yes |
| 12 | TCDD (*Solvin et al., 2023*) | 2015 | 1,568 | PCG | Yes |
| 13 | Digiscope (*Kavitha & Renumadhavi, 2022*) | 2017 | 29 | PCG | Yes |

datasets using ML have evolved specifically in terms of complexity and diversity. Collecting data sets for heart disease poses significant challenges due to a variety of subject constraints, such as infants and toddlers' age, their tendency to cry and breathe heavily, the presence of external noise, and the inclusion of pregnant women. Researchers gather these data sets using various methods, such as CT images, ECG recordings, TTEs, phonocardiograms (PCGs), and ultrasound images and videos.

Understanding the complexity of CHD often depends on the quality and source of the research data. When studying this topic, researchers typically opt for one of two approaches: using publicly available datasets or collecting data straight a way from hospitals or medical research centers. This section offers a comprehensive overview of these datasets, highlighting their evolving nature and the crucial role they play in advancing our understanding of CHD.

Researchers investigating the prediction and detection of CHD employ a variety of methodologies, leveraging the capabilities of ML models in multiple modalities. Some researchers (*Boneva et al., 2001*; *Rosamond et al., 2007*; *Luo et al., 2017*) focus on 3D cardiac MRIs of the heart, taking advantage of the intricate details provided by this advanced imaging technology. Others explore high-quality 2D EKG images, with detailed visualizations for their predictive analyzes. Another cohort of researchers examines images of singleton fetuses or screening videos of pregnant individuals to predict CHDs in unborn infants. In addition, some researchers focus on ECG signal recordings or heart sounds, using these physiological markers to develop predictive models. Lastly, other researchers use classical tabular data with a range of characteristics related to cardiovascular health to discern patterns and develop predictive algorithms. This multidisciplinary approach emphasizes the versatility of ML in healthcare care, taking advantage of various data sources to improve CHD prediction and early detection. All data sets and modalities used for the recognition of CHD are listed in Table 1.

### ZCHSound (*Liu et al., 2022*)

Data collection was carried out at the Children's Hospital of the School of Medicine of Zhejiang University (ZU) and numerous affiliated institutions in China. The ZU Kids Hospital is a well-known medical facility in the country that specializes in providing comprehensive healthcare services to children. All participants who willingly chose to participate in the study had explicit approval from their guardians/parents.

The heart sound data were acquired using an intelligent stethoscope (frequency: 8,000 Hz). Throughout the procedure, highly trained physicians specializing in specimen collection and clinical evaluations performed the tasks to ensure the precision of the data. The stethoscope was placed near the left border of the sternum, between the second and third ribs, to perform cardiac auscultation.

For each participant, the duration of each recording of heart sounds varied between 11 and 30 s. Gathering cardiac sounds from infants and young people has specific challenges, as the presence of crying, coughing, intestinal motility, and physical exertion could produce additional noise. As a result, the heart sound data that is gathered may contain a substantial quantity of low-quality data that is contaminated with noise. In order to streamline the process of choosing suitable data for analysis, annotations are provided indicating the quality of the data. Skilled physicians have examined the heart sound data that was gathered.

According to the information we have, this is the largest and most authentic database with CHD PCG data. The heart sound data set includes 941 participants, each of whom has one audio recording. Each audio recording is roughly 20 s long, with a total duration of 5 h. The data set comprises 473 women and 468 men. The study includes 533 individuals without heart disease as a control group. However, other individuals have received diagnoses for ASD, PDA, PFO, and VSD. The number of cases for each disease is 119, 32, 70, and 187, respectively. The candidate's ages range from 2 days to 14 years.

### The CirCor DigiScope dataset (TCDD) (*Solvin et al., 2023*)

This data set contains heart sound data collected from 2014 to 2015. The data set included 5,282 heart sounds recorded from 1,568 patients, with 787 males and 781 females. The participants' ages ranged from three days to thirty years. The sounds were recorded with a Littmann 3,200 stethoscope for a duration of 30 s. Heart sound data are evaluated for quality and murmurs. Two separate cardiac physiologists annotate the data to ensure precision. The samples obtained were segmented with three automated algorithms. Then, two cardiac physiologists independently assessed the segmentation findings and evaluated the outcomes that were particular to each algorithm.

### Heart sounds Shenzhen corpus (HSS) (*Qiao et al., 2022*)

The data set comprises heart sound data obtained from 170 participants. Heart sounds were captured from four standard auscultation sites using a stethoscope and Bluetooth 4.0 technology. Each location was recorded for a duration of 30 s. The data were analyzed by EKG, which used the area ratio of the mitral valve and the tricuspid valve to predict regurgitation. The severity of regurgitation was classified as mild, moderate, or severe. The heart sound data set is categorized into three groups, including severe, mild, and moderate.

### DigiScope2017 (*Kavitha & Renumadhavi, 2022*)

The heart sound data were obtained from a sample of 29 healthy youngsters, whose ages ranged from 6 months to 17 years. Heart sounds were obtained at the Royal Portuguese Hospital, with a recording time ranging from 2 to 20 s. The heart sounds were recorded at a frequency of 4,000 Hz while the stethoscope was placed in the mitral position. A cardiovascular physician manually identified the start and end times of S1 and S2 using specialized software to analyze heart sound data.

### Heart ventricles segmentation in MR (*Wolterink et al., 2016*)

Researchers from Boston Children's Hospital and MIT worked together to collect the heart ventricles segmentation in MR (HVSMR) data set. The HVSMR data set consists of 3D CMR images captured during routine clinical practice. This data set is a valuable resource for studying CHD, encompassing cases with a diverse range of cardiac abnormalities, including those that have undergone interventions.

The imaging was performed using a 1.5T magnetic resonance imaging scanner, employing a steady-state free precession pulse sequence in axial view. During image acquisition, respiratory navigation and ECG techniques were employed to mitigate motion artifacts caused by cardiac and respiratory activity. The data set includes manual segmentations of the blood pool and ventricular myocardium, meticulously performed and validated by trained raters and clinical experts.

Segmentation was performed primarily in a short-axis view and subsequently transformed back to the original image space for analysis. Segmentation of the blood pool includes different heart structures, such as the atria, ventricles, aorta, pulmonary veins, and vena cavae. However, longer vessel segments are left out to make 3D modelling of the heart surface better for planning surgery. This dataset, with openly available cropped and short-axis images for each subject, serves as a critical resource to advance automated segmentation algorithms and facilitate research in the field of CHD.

The HVSMR-2.0 dataset represents a significant contribution to the field of CHD research, addressing the critical need for comprehensive CMR datasets with manual segmentation masks. This data set includes 60 CMR scans accompanied by detailed manual segmentation masks of the four cardiac chambers and four great vessels.

The images encompass a diverse spectrum of heart defects and surgical interventions, facilitating the advancement of automated segmentation algorithms and the development of innovative tools for surgical planning and simulation. In addition, the data set includes masks that delineate the required and optional extents of the great vessels, enhancing the validity of comparative analyses among algorithms. HVSMR-2.0 sets the stage for the development of clinically relevant tools that can significantly impact the care and treatment of patients with CHD.

### Physionet (*Garcia-Canadilla et al., 2020*)

This data set is collected from pregnant women while recognizing the importance of detecting CHD during pregnancy. This disease is also very complex and life-threatening, so early detection is necessary. Data collected for research are in the form of electrocardiogram

(ECG) signals from pregnant women. For publicly available data, the resource PhysioBank is used.

The database collected data is both normal and abnormal women. The ECG signals are collected over an extensive monitoring period of 20 h. The signal is recorded at a frequency of 250 Hz. The research data set consists of two groups, both pregnant women. The first group has 15 women, and the next has 18. The women in the first group have children who have been diagnosed with CHD while the children in the second group are healthy. A subset of pregnant women are identified within the first group who have severe CHD.

### Doppler TTE images (*Hrusca et al., 2016*)

This data set included 2D and Doppler TTE scans of 1,932 children, collected at Beijing Children's Hospital (BCH), collected between 2018 and 2022. The initial data set consisted of 1,080 children, 823 of whom were healthy controls, 209 had ASD, and 276 had VSD. The second data set consisted of 624 children, with 432 healthy controls, 83 with ASDs, and 109 with VSDs. These data sets were obtained from several sonographers at BCH.

Patients in the heart center provided all samples for this study. The TTE data from a group of 1,932 people who did not have any structural problems with their hearts was used as healthy control data after the assessment. Individuals diagnosed with ASD or VSD were selected as positive instances. A minimum of two highly experienced senior sonographers, a chief physician with over 15 years of experience and over 150 thousand US examinations, or a final intraoperative diagnosis determined the status of all subjects. The data acquisition equipment used in this study was the PHILIPS iE 33, iE Elite, and EPIQ 7C US machines from Philips Electronics Nederland.

### DICOM X-rays dataset (*Gerke, Minssen & Cohen, 2020*)

The DICOM data set comprises 828 chest radiograph files obtained from children, classified into four groups: atrial septal defect (194), ventricular septal defect (210), patent ductus arteriosus (216) and a control group consisting of individuals with typical characteristics (208). A cardiac ultrasound report of the corresponding child, which confirms the absence of additional cardiac or pulmonary disorders, follows each chest radiograph that corresponds to a particular heart problem. Three experts in US imaging evaluated cardiac US reports.

Cardiologists compiled data on paediatric patients with CHD who had hospitalizations and underwent regular post-treatment evaluations, which included chest radiographs and cardiac ultrasound. The time difference between the cardiac US report and the chest radiograph is no more than 3 days. Three specialized cardiac ultrasound physicians examined the cardiac ultrasound reports and confirmed the diagnosis of CHD after initially storing chest radiograph images in DICOM format.

### C-XRAY (*Seah et al., 2019*)

A total of 103,489 frontal chest radiographs were obtained from the author's institution. These radiographs were obtained between January 1, 2007, and December 31, 2016, and belonged to 46,712 different patients. All radiographs recovered were included in the study data set, without applying any exclusion criteria. The REASON Cohort Discovery

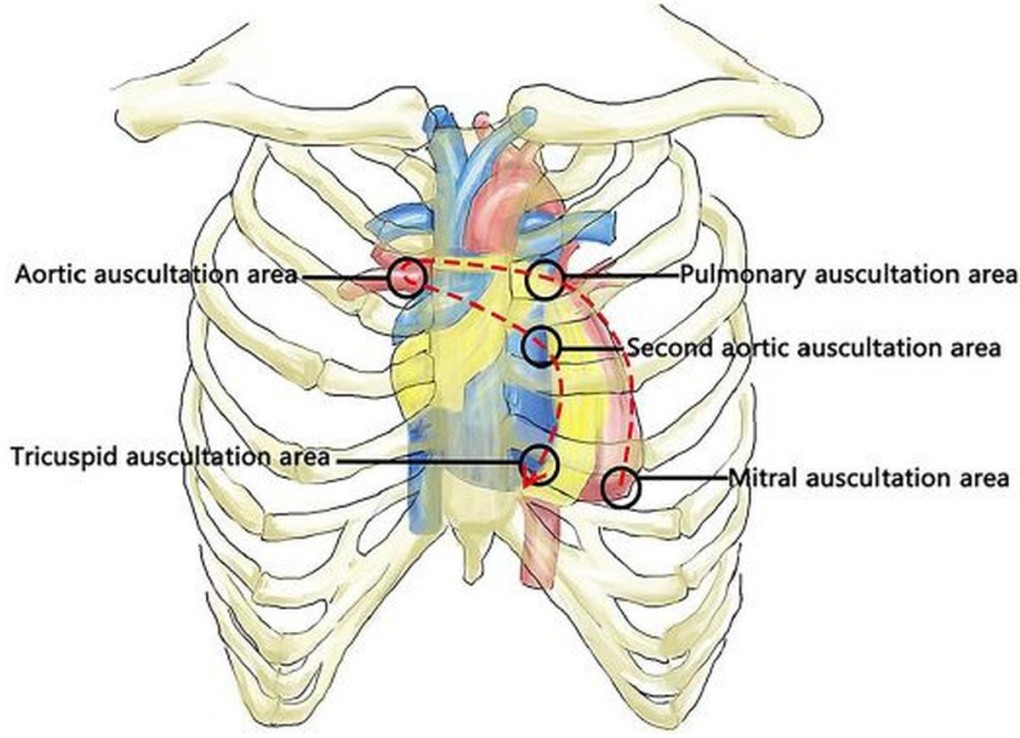

**Figure 6** Diagram illustrating the sequence of auscultation for heart sounds (*Chen et al., 2024*) .

Tool found 7,390 radiographs from 5,232 different patients with a corresponding BNP value within 36 h of image capture. These radiographs make up the labelled data set. The unlabelled data set consisted of 96,099 radiographs that did not have a corresponding BNP result. The radiographs were resized to a resolution of $128 \times 128$ pixels.

### Chongqing Medical University dataset (*Chen et al., 2024*)

The Chongqing Medical University dataset (CMUD) included 475 patients with CHD, aged $4.2 \pm 3.1$ years, and 409 patients without structural heart abnormalities, aged $5.3 \pm 2.9$ years. A diagram illustrating the sequence of auscultation for heart sounds is shown in Fig. 6. These individuals were hospitalized in the Department of Cardiology at the Children's Hospital of the University of Chongqing, China. The data set did not include participants over 14, those with inherited disorders, or those unable to complete the collection of heart sounds. Children who were diagnosed with CHD through EKG were classified into the CHD group, while children without CHD were classified into the control group. The CHD group was separated into subgroups based on heart disease, including ASD, VSD, PDA, and combined CHD. These subgroups represent the most frequently observed types of CHD with a left-to-right shunt.

The collection of all heart sounds was performed using the 3M Littmann Electronic Stethoscope. Information was collected from five specific regions of the body. These locations are the mitral auscultation area, the pulmonary auscultation area, the aortic auscultation area, the second aortic auscultation area, and the tricuspid auscultation area.

Each region was measured for an average period of 10 s while the individual was lying on their back, with a 2–3 s gap between neighbouring regions. For a duration of 60 s, one recording consisted of entire heart sounds from five auscultation regions. All records were transmitted by Bluetooth to the computer. Two cardiologist physicians with more than a decade of clinical experience performed the hearing diagnosis. The age, gender, height, weight, EKG diagnostics, and other pertinent characteristics of each participant were documented.

### POX dataset (*Roberts et al., 2012*)

The POX dataset is collected over a 2-year study period. The hospital recorded a total of 44,147 live births. Most newborns were born at full term, which means that they were born between 37 and 40 weeks of gestation. The median birth weight of the newborns was 3,420 grams. A total of 498 infants with CHD were first identified, 27 were detected by POX screening, and 471 by cardiac auscultation. This resulted in an overall screening rate of 1.13% among the 44,147 live births. Of the total number of cases, 458 newborns were verified using ECG, resulting in an overall diagnosis rate of 92% for CHD using this imaging technique. This includes 253 male infants and 245 female infants. The predominant forms of CHD were patent ductus arteriosus (PDA), accounting for 34.3% of cases, atrial septal defect (ASD), accounting for 20.5% of cases, ventricular septal defect (VSD), accounting for 8.3% of cases, and combined problems accounting for 34.5% of cases.

The rare CHDs that were observed included coarctation of the aorta (COA), coronary artery anomalies (CTA), partial anomalous pulmonary venous connection (PAPVC), total anomalous pulmonary venous connection (TAPVC), transposition of the great arteries (TGA), and fallot tetralogy (TOF). These conditions accounted for a total of 2.2% of the cases. Of all instances, 74% were attributed to mild CHD, while intermediate CHD accounted for 15.3% and severe CHD accounted for 10.7%. Of the 458 confirmed participants with CHD, the majority of 438 cases were recognized alone using cardiac auscultation, while just 20 instances were identified by pulse oximetry (POX) screening alone. However, no CHD was detected by auscultation and POX screening. The POX screening method had a rather poor accuracy of 74.07% in terms of positive predictive value (PPV). Using only auscultation yielded a high positive predictive value (PPV) and a negative predictive value (NPV) of 92. 99% and 99. 95%, respectively. However, the use of pulse oximetry (POX) further enhanced the screening performance, resulting in a 100% NPV.

### UCI dataset (*Arooj et al., 2022*)

The heart patient data set used in the proposed technique was obtained from the publicly available repository of the University of California (UCI, Irvine CA), which can be accessed on the Kaggle website (https://www.kaggle.com/datasets/redwankarimsony/heart-diseasedata). The data set was accessed on June 15, 2022. The data set obtained through Kaggle comprised data from 1,050 patients, covering 76 variables. Among the total of 76 qualities, only 14 were used in the prediction of heart disease. This is because the other attributes have a lower impact on the disease compared to these attributes.

Prior to categorization, the data set undergoes a process of cleaning and filtering to remove missing or redundant variables. The data set was partitioned into training and testing datasets, with 80% and 20% of the samples, respectively, chosen at random. Among the 1,025 patient records, 820 were allocated for training purposes, while the remaining 205 samples were reserved for testing.

# KEY CHALLENGES AND LIMITATIONS

The integration of ML and DL-based methods into pediatric cardiology presents a number of challenges. In various areas of paediatric cardiology, these AI-based methods are beneficial. Some of these methods include examination and clinical diagnosis, image processing of foetal cardiology, risk stratification and prognosis, planning of cardiac interventions, and lastly, precision cardiology. ML-based algorithms are promising tools for diagnosing both critical and noncritical CHDs. However, we believe that significant effort remains in both the ML and health sector domains. We are far from the point where ML-based algorithms can diagnose a complex CHD with computers. There are numerous reasons for this, but we aim to encapsulate these challenges in this section of our article.

## Limited Access to high-quality data

The effectiveness of most AI systems is greatly influenced by the quality and volume of training data. Furthermore, the scanned data exhibit considerable quality variations in addition to their high cost. The diversity in data quality can hinder the development of a universal network and present a substantial obstacle to the commercialisation of AI-driven solutions. The datasets available to ML and DL experts are very limited for CHD recognition. To train AI-based models, we need sufficient training and testing data. To assess and, along with that, investigate inherent biases and overfitting, we need sufficient data. In addition, the heterogeneity in cardiac anatomy and the variety of individual disease entities make the incorporation of DL-based solutions into this field difficult. To address this limitation, data from all sources were needed, which is not possible in the near future. Researchers have not yet tried transfer learning as another strategy to address this issue.

## Data acquisition problems

Currently, the healthcare industry is collecting data from multiple facilities and patients in order to identify CHD. Using these data effectively, doctors can predict more advanced treatment options and improve the whole healthcare delivery system for this disease. Unfortunately, the duration of data collection for CHD is very short. Data collection is necessary immediately after birth. With the passage of time, the disease becomes more dangerous. Most people scrutinise their children immediately after birth. For proper treatment, it is necessary to know the disease in the initial days.

Imaging in the paediatric population presents a significant challenge due to their smaller size and unique movements during the image acquisition process. This also leads to higher motion artefacts during the acquisition process. This situation leads to a technical challenge that requires a comparatively higher spatial resolution, specifically in acquiring the MRI signal (*Kholmatova, Kharkova & Grjibovski, 2016*). This scenario leads to a situation where doctors and patients may hesitate to use ML/DL to replace the currently available protocols.

## Data privacy concerns

When patient data is collected for research or industrial use, proper documentation is performed. The applicants sign consent forms, confirming their consent to the use of data for research or commercial purposes. This practice is also time-consuming, and most participants hesitate to participate in such activities.

The field of AI is evolving, but there are also growing ethical concerns. Several considerations include obtaining authorization for data access, ensuring data security and privacy, addressing fairness and biases in algorithms, and promoting openness (*Ng et al., 2022*). The ethical difficulties that can arise from the use of AI in the health industry are not adequately addressed by existing laws and regulations. This will take time to develop the proper regulations. However, we believe that AI is rapidly expanding, and with this growth, we can explore laws that will ensure algorithmic transparency and data privacy.

## Limitations in implementing ML

View classification is an essential step in developing a completely automated system. Currently, there are other obstacles to building a perspective. For example, a classification model is used to compare paediatric EKG. There are various changes in anatomy, size, structure, and perspectives between the two (*Gearhart et al., 2022*). Mathematical models, along with ML and AI technologies, show promise in predicting outcomes in the paediatric cardiac ICU by analysing continuously recorded physiological information. As the success of predictive systems relies on the accuracy of the results they predict, it is essential to consistently create acceptable endpoints as this technology advances. Although ML can provide estimates for the likelihood of certain outcomes, it cannot beat the clinical judgement of physicians when it comes to choosing the most appropriate treatment course based on clinical circumstances to prevent clinical decompensation (*Sakai et al., 2022*).

## Annotated data scarcity

In order to assess a collection of CHD algorithms, it is necessary to have accurate and reliable reference data. Ground truth data can be obtained in several ways, but the process of collecting and annotating these data is challenging, resulting in the presence of inaccuracies and noisy information in most ground truth annotations. Potential errors may be attributed to the incorrect behavior of participants during the acquisition process. The acquisition sensor might also affect the quality of the data.

When dealing with complicated acquisition situations, a viable option to train and assess a ML-based CHD framework is to utilize synthetic datasets. When using synthetic datasets, the likelihood of mistakes is lower compared to those obtained in more realistic scenarios. Unfortunately, the shape and geometry of the heart are very complex. Even medical professionals are unable to understand the full development of the heart and its abnormalities. Therefore, the literature lacks synthetic datasets for CHD recognition that ML/DL algorithms can use.

The annotation of data by medical professionals and neonatologists is another problem. Humans do most of the annotation and labelling. One of the oldest methods for generating ground truth data is a practitioner assigning a label based on their personal perception of

CHD. Labelling small data sets is easy with this strategy; however, it becomes inappropriate for large databases since the probability of human error is high. Sometimes, the perspectives of doctors differ, leading to confusion in labelling and annotation. In a nutshell, the creation of ground-truth data is also a major problem to be explored. Due to this, very few databases are reported in the literature.

## RESEARCH GAPS AND FUTURE DIRECTIONS

CHD is not a mature research area, not only for computer vision and ML experts, but also for medical practitioners (*Mullen et al., 2021*; *Arooj et al., 2022*; *Q-m et al., 2014*). The area remains unexplored for many complex reasons. We discussed some possible reasons in 'Key Challenges and Limitations' as computer vision and ML experts.

Hospitals generate large amounts of data, encompassing clinical information, genomic data, and data from electronic health records. The continual progress in big data is critical in healthcare administration because it allows the analysis of large datasets to improve illness treatment, determine appropriate therapeutic dosages, and make predictions (*Khan, Ahmad & Uddin, 2023*). Healthcare produces a large amount of data, but most of it remains untapped due to challenges in storing, maintaining, and analysing complex datasets that often involve multidimensional and nonlinear relationships between variables.

The use of these datasets, particularly in rare conditions such as CHD, with AI predictive models can be beneficial in identifying individuals who are at risk of having children with CHD (*Khan, Ahmad & Uddin, 2023*; *Moonesinghe et al., 2013*; *Min, Yu & Wang, 2019*; *Olive & Owens, 2018*). Table 2 presents a summary of all the ML and DL-based methods for the recognition of CHD. Similarly, all data sets used for CHD recognition are also reported in Table 1.

It is very challenging to organise all the approaches that use ML for CHD recognition into a single taxonomy. The challenge lies not in the abundance of methods proposed by ML experts, but in the diversity of data types used for problem analysis. For example, the techniques used to analyse ECG signals differ from those used to examine MRI images. Therefore, we are not adapting and do not follow a specific taxonomy. We will endeavour to associate each model with its corresponding implementation. Establishing connections between different methods for CHD recognition is a challenge. In the following paragraphs, we provide a thorough analysis of these methods and scrutinise the scholarly articles that specifically focus on them. In addition, we offer a thorough assessment of the advantages and disadvantages associated with each method. Figure 7 summarizes how CHD has been addressed by researchers.

### Conventional ML methods

Conventional ML techniques have employed a variety of methods to address the recognition and classification of CHD. Feature engineering in traditional ML (TML) methods is a more intricate and labour-intensive process. TML comprises several discrete stages, including pre-processing, feature extraction, feature selection, and classification.

The study by *Huang et al. (2022)* uses cardiac signals to categorise CHD. The authors of this study have identified numerous diverse and comprehensive features. In a random

**Table 2  Year-wise development of CHD from 2016 to 2024.**

| Year | Approaches | Method | Data type | Database | Metrics |
|---|---|---|---|---|---|
| | Liu et al. (*Chen et al., 2024*) | DLM | PCG | CHDdECG | $CHD_{acc}$, Sen., Spec. |
| | Qiao et al. (*Pachiyannan et al., 2024*) | DLM | EKG | PhysioNet | $CHD_{acc}$, Sen., Spec. |
| | Cheng et al. (*Hrusca et al., 2016*) | DLM | ECG | Doppler | AUC, $CHD_{acc}$ |
| 2024 | Li et al. (*Gerke, Minssen & Cohen, 2020*) | DLM | X-rays | DICOM | $CHD_{acc}$, ROC, and $CHD_{mat}$ |
| | Chen et al. (*Q-m et al., 2014*) | DLM | ECG | CHDdECG | $CHD_{acc}$, Sen., Spec. |
| | Cornforth et al. (*Cornforth & Jelinek, 2016*) | DLM | ECG | PhysioNet | $CHD_{acc}$, Sen., Spec. |
| | Prabu et al. (*Garcia-Canadilla et al., 2020*) | DLM | ECG | PhysioNet | Sen., Spec. |
| | Xu et al. (*Huang et al., 2022*) | TMLD | PCG | – | $CHD_{acc}$ |
| 2023 | Ng et al. (*Arafati et al., 2019*) | TMLD | CT images | PhysioNet | Sen., Spec. |
| | Solven et al. (*Jahmunah et al., 2024*) | TMLD | ECG | – | Sen., Spec. |
| | Kavitha et al. (*Jia et al., 2024*) | DLM | ultra sound | – | $CHD_{acc}$, RMSE, $CHD_{mat}$ |
| | Gearhart et al. (*Gearhart et al., 2022*) | DLM | ECG | – | $CHD_{mat}$ |
| | Van et al. (*Lv et al., 2021*) | DLM | EKG | – | $CHD_{acc}$, Sen., Spec. |
| | Xu et al. (*Olive & Owens, 2018*) | DLM | ECG | – | $CHD_{acc}$, Sen., Spec. |
| 2022 | Chang et al. (*Chang et al., 2022*) | Random-forest | ECG | unkown | $CHD_{mat}$ |
| | Arooj et al. (*Arooj et al., 2022*) | CNNs | EKG | UCI | $CHD_{acc}$, $CHD_{mat}$ |
| | Vayadande et al. (*Vayadande et al., 2022*) | TML | EKG | UCI | $CHD_{acc}$, $CHD_{mat}$ |
| | Botros et al. (*Botros, Mourad-Chehade & Laplanche, 2022*) | CNNs and TML | ECG | Physionet | $CHD_{acc}$, Sen., Spec. |
| | Van et al. (*Thomford et al., 2020*) | CNNs | PCG | – | $CHD_{acc}$, Sen., Spec. |
| | Hoodatasethoy et al. (*Hoodbhoy et al., 2021*) | review paper | – | – | – |
| 2021 | Morris et al. (*Morris & Lopez, 2021*) | CNNs | ECG | – | $CHD_{acc}$ |
| | Eltrass et al. (*Eltrass, Tayel & Ammar, 2021*) | CNNs | ECG | MIT, BIDMC | $CHD_{acc}$, Sen., Spec. |
| | Hussain et al. (*Hussain et al., 2021*) | TML | ECG | – | ROC, AUC, Sen., Spec. |
| | Thomfard et al. (*Zhang et al., 2015*) | review article | – | – | – |
| | Ning et al. (*Ning et al., 2020*) | CNNs, RCNNs | ECG | Data from internet | Sen., Spec. |
| | Chang et al. (*Chang Junior et al., 2020*) | TML | ECG | – | Sen., Spec. |
| 2020 | Jonnavithula et al. (*Jonnavithula et al., 2020*) | TML | EKG | – | $CHD_{acc}$ |
| | Diller et al. (*Diller et al., 2020*) | PG-GAN,U-Net | MRI | – | $CHD_{acc}$ |
| | Rani et al. (*Rani & Masood, 2020*) | TML | ECG | Shanxi CHD | $CHD_{acc}$ |
| | Porumb et al. (*Porumb et al., 2020*) | CNNs | EKG | MIT, Physionet | $CHD_{acc}$, Sen., Spec. |
| | Garcia et al. (*Garcia-Canadilla et al., 2020*) | review article | – | – | – |
| | Acharya et al. (*Acharya et al., 2019*) | CNNs | EKG | PhysioBank | $CHD_{acc}$, Sen., Spec. |
| | Bhurane et al. (*Bhurane et al., 2019*) | TML, Wavelet | ECG | Physionet, BIDMC | $CHD_{acc}$, Sen., Spec. |
| | Wang et al. (*Wang et al., 2019*) | deep ensemble | EKG | BID-MC, MIT-BIH | $CHD_{acc}$ |
| 2019 | Combi et al. (*Combi & Pozzi, 2019*) | | review article | | |
| | Seah et al. (*Seah et al., 2019*) | Neural netowrk | X-rays | CXRAY | AUC, ROC |
| | Isler et al. (*Isler et al., 2019*) | LSTM, TML | ECG | Physionet | $CHD_{acc}$, Sen., Spec. |
| | Liu et al. (*Liu & Kim, 2018*) | LSTM | ECG | Physionet | $CHD_{acc}$ |
| 2018 | Kaouter et al. (*Masetic & Subasi, 2016*) | CNNs | ECG | MIT-BIH, BIDMC | $CHD_{acc}$ |
| | Li et al. (*Kaouter et al., 2019*) | CNNs | DDM | Physionet | $CHD_{acc}$ |

**Table 2** (*continued*)

| Year | Approaches | Method | Data type | Database | Metrics |
|------|-----------|--------|-----------|----------|---------|
| | Li et al. (*Li et al., 2017*) | BPNN | ECG | – | $CHD_{acc}$, Sen., Spec. |
| **2017** | Meystre et al. (*Meystre et al., 2017*) | TML | notes | – | $CHD_{acc}$, Sen., Spec. |
| | Masetic et al. (*Masetic & Subasi, 2016*) | TML | ECG | Physionet | Sen., Spec., ROC |
| | Chen et al. (*Li et al., 2018*) | Auto-encoder | ECG | Physionet | $CHD_{acc}$ |
| **2016** | Luo et al. (*Chen et al., 2017*) | TML | ECG | – | $CHD_{acc}$ |
| | Pace et al. (*Pace et al., 2015*) | TML | EKG | – | $CHD_{acc}$ |

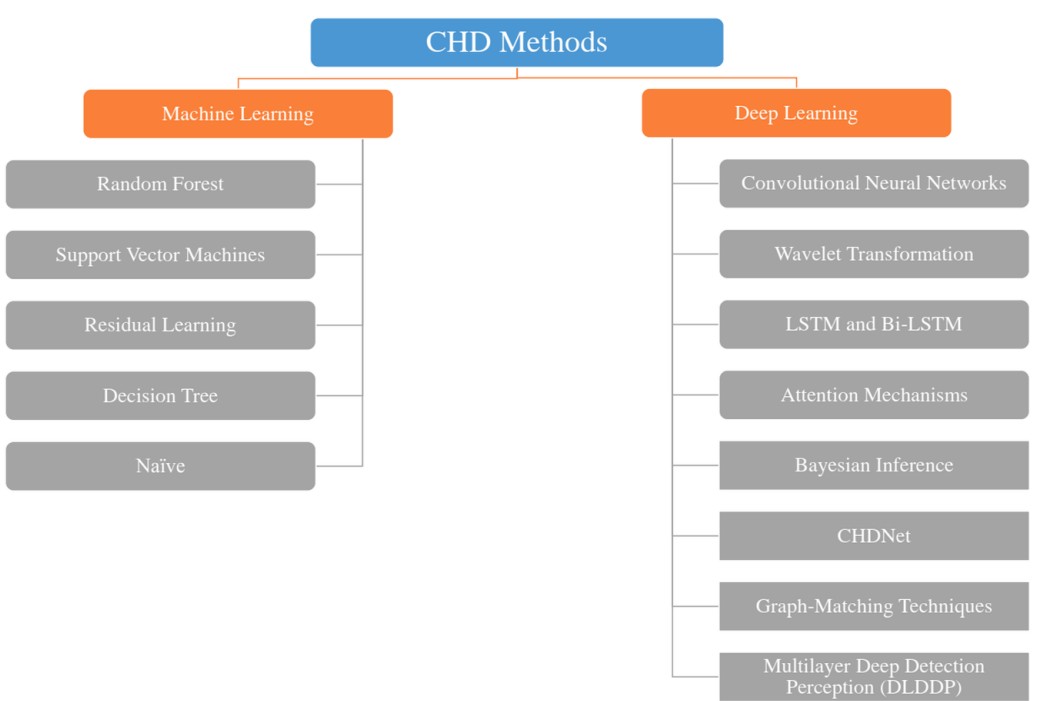

**Figure 7   Overview of machine learning and deep learning techniques for CHD detection.** This diagram illustrates key methods such as Random Forest, Support Vector Machines, and Decision Trees within ML, and Convolutional Neural Networks, LSTM, and Attention Mechanisms within DL, highlighting their respective roles in CHD detection and classification.

forest, the authors used information from the frequency domain and wavelets to build conventional classifiers. The authors of the article assert that state-of-the-art (SOA) datasets have significantly improved outcomes. The authors conducted a cohort study from May 2020 to December 2021. The researchers simulated the anatomical structures of the hearts and major blood vessels of 29 children with congenital heart defects. Among these 29 individuals, 26 were infants who were born at the clinic of St. Petersburg State Paediatric Medical University. The patients were in two groups. Consents forms were taken from the parents of the kids for this study. They were informed about this examination and treatment for scientific research.

Some pre-processing and post-processing methods have been applied to the collected data, which are listed in *Chang (2019)*, *Sreedhar et al. (2005)*, *Jahmunah et al. (2024)*,

respectively. Similarly, Ng et al. (*Arafati et al., 2019*) designed a framework for complex CHD classification by leveraging retinal images. The authors of this article have introduced a very innovative and distinctive feature extraction method. This work combines both colour and texture-based features. Then, the authors apply conventional ML (support vector machines) to the extracted features for risk classification. The authors compare the results with previously implemented frameworks and report improvements.

*Pachiyannan et al. (2024)* proposed a residual learning-based system they called RLDS. The method employs a residual learning strategy, which is designed for cases of fetal CHD. The RLDS extracts distinctive features from images of EKG. The RLDS generates attention maps, which assign importance scores to each feature. This helps in enhancing the understanding of the diagnostic process for medical professionals. The authors reported better accuracy on SOA dataset.

As stated by the authors in *Bhurane et al. (2019)*, the ECG is not reliable for diagnosing CHD for medical practitioners. Analysing the ECG signal to detect CHD requires manual effort and the ability to identify subtle abnormalities in the heart's electrical activity, which demands expertise and skill. The authors suggested using ML to expedite and enhance the identification of various ECG signal abnormalities associated with CHD. The authors propose an automated method for diagnosing CHD by analysing ECG signals. The proposed methodology was assessed using four separate collections of CHD (ECG signals) datasets. The experiments were conducted using short (2 s) ECG segments. Five unique attributes (fuzzy entropy, Renyi entropy, Higuchi's fractal dimension, Kraskov entropy, and energy) were obtained through the wavelet decomposition of ECG segments using frequency-localized filter banks. They utilised a quadratic support vector machine for both training and classification purposes. They conducted an evaluation using a 10-fold cross-validation technique. Accuracy, Sensitivity (Sen), Specificity (Spec). were calculated for all four datasets. Hospitals can implement the technique to streamline the diagnosis of CHD. However, we contend that the current implementation is not feasible. Developing this area to a more advanced stage will require a significant amount of time.

The study proposed in *Hussain et al. (2021)* involved the ranking of multimodal features collected from people with CHD and normal sinus rhythm. Using the values of empirical receiver operating characteristics, the proposed method classified the features into five groups, ranging from 1 to 5. Instead of utilising all multimodal features, authors employ only the most highly ranked aspects to detect CHD and normal patients. The study employed resilient ML methods such as decision tree, naïve Bayes, SVM (Gaussian, RBF, and polynomial). The performance was assessed using sensitivity, specificity, positive predictive value (PPV), negative predictive value (NPV), accuracy, false positive rate, and AUC-ROC.

The SVM-Gaussian model achieved the best detection performance in terms of accuracy and AUC when using all multimodal features. The model had a sensitivity of 93.06%, specificity of 81.82%, an accuracy of 88.79%, and an AUC of 0.95. The highest performance was achieved using SVM Gaussian with the top five ranked features, resulting in an accuracy of 84.48% and an AUC of 0.86. When using Decision Tree and Naïve Bayes with the top nine ranked features, the accuracy and AUC remained the same at 84.48% and 0.88, respectively.

Lastly, when using SVM polynomial with the last thirteen ranked features, the accuracy dropped to 80.17% and the AUC decreased to 0.84. The results suggest that the proposed method, which includes feature rating, can be highly beneficial for automatically detecting patients with congestive heart failure. This approach can also greatly assist clinicians and physicians in making informed decisions to reduce the death rate.

## Deep learning based methods

The advancement of DL algorithms has greatly revolutionised computer vision techniques. In the past, computer vision primarily relied on manually crafted features and methods to interpret visual data, sometimes leading to limited performance in complex tasks. However, the implementation of DL, particularly convolutional neural networks (CNNs), has entirely revolutionised this field of research. DL models have the capability to independently acquire hierarchical representations from raw data, hence improving their ability to effectively capture intricate patterns and features. The implementation of this novel methodology has significantly enhanced the accuracy, robustness, and scalability of computer vision systems across various domains, such as object detection, image categorization, facial identification, and medical imaging.

DL has greatly enhanced computer vision by leveraging enormous amounts of annotated data and high-performance computational resources. This has led to unprecedented levels of performance and generated potential for a wide range of innovative applications and solutions. Similar to other methods, DL-based algorithms for CHD recognition have also shown increased performance compared to previous approaches using SOA datasets. Table 2 reveals that the majority of algorithms employed in current research are based on DLM.

These DLM, specifically those based on CNNs, outperform conventional feature-based methods. The shift from conventional ML to DL methods has mitigated many drawbacks and serious limitations of traditional approaches. A DL-based method is proposed in *Chen et al. (2024)*. The proposed method, which the authors named CHDdECG, utilises DL to diagnose CHD by extracting features from paediatric ECG and wavelet transformation characteristics. The authors then combine these features with important human-concept variables. CHDdECG was tested on a dataset of 65,869 instances and achieved a ROC-AUC of 0.915 and a specificity of 0.881 on a real-world test set consisting of 12,000 patients. In addition, the proposed algorithm was tested on two separate external datasets consisting of 7,137 and 8,121 instances; the overall ROC-AUC values were 0.917 and 0.907, respectively. The specificities achieved were 0.937 and 0.907. CHDdECG outperformed cardiologists in detecting CHD, as indicated by the comparison of their performance. The extracted ECG features' importance scores suggested that they have a greater impact on CHD detection compared to human-concept features. This implies that CHDdECG may possess knowledge that goes beyond human understanding. The proposed work has a direct influence on the identification of CHD using paediatric ECG. In this study, the authors make use of the PCG signals. One of the main drawbacks of this method is that the proposed framework is limited to one type of CHD. It also depends on the high-quality heart sound data.

Early detection and diagnosis are crucial for the effective treatment of CHD, a complex medical issue. This is because CHD can manifest in various ways and have mild symptoms that are present from birth. The research article proposed in *Pachiyannan et al. (2024)* presents a revolutionary healthcare application called the ML-based CHD Prediction Method (ML-CHDPM). The purpose of this method is to overcome the difficulties and speed up the accurate detection and categorization of CHD in pregnant women. The ML-CHDPM model utilises cutting-edge ML techniques to classify cases of CHD, using relevant clinical and demographic parameters. The model has been trained on a rich dataset, enabling it to accurately grasp subtle patterns and correlations, leading to precise predictions and classifications. The assessment of the model's performance includes sensitivity, specificity, accuracy, and the area under the receiver operating characteristic curve. The findings highlight the ML-CHDPM's exceptional performance in six key metrics: accuracy, precision, recall, specificity, false positive rate, and false negative rate. The approach attains an average accuracy rate of 94.28%, precision of 87.54%, recall rate of 96.25%, specificity rate of 91.74%, false positive rate of 8.26%, and false negative rate of 3.75%. This research represents a notable advancement in the field of ECG signal processing, utilising sophisticated ML algorithms to enable the early identification and diagnosis of medical conditions, with a specific focus on pregnant women.

The research undertaken by *Cheng et al. (2024)* includes examining the 2D and Doppler TTEs of children from two different clinical groups at BCH. The data was gathered from 2018 to 2022. A DL framework was developed with the purpose of identifying cardiac views, integrating data from various perspectives and modalities, visualising the high-risk region, and estimating the probability of an individual being healthy or having an ASD or a VSD. The DL model attained a mean accuracy of 0.989 for view classification. The CHD screening approach employs both 2D and Doppler TTEs, which capture 5 distinct perspectives. The model attained a mean AUC of 0.996 and an accuracy of 0.994 when evaluated within the same centre. When evaluated in several centres, the model had a mean AUC of 0.990 and an accuracy of 0.993.

The model achieved a mean accuracy of 0.991 and 0.986 for within-centre and cross-centre evaluation, respectively, in classifying healthy, ASD, and VSD. The DL models that combine many modalities and scanning viewpoints achieved better performance, approaching that of professional sonographers. By incorporating various perspectives and methods of transthoracic EKG TTEs into the model, it becomes possible to accurately identify children with CHD in a way that does not require invasive procedures. This suggests that there is potential to improve the performance of CHD detection and simplify the screening process.

According to the study in *Tan et al. (2023)*, an AI-driven CHD diagnosis network called CHDNet is created. It is a binary classification model that looks at EKG videos to decide if they show heart problems or not. According to the authors, the CHDNets have demonstrated comparable or superior performance to medical specialists. The authors suggest two approaches: Bayesian inference (BI) and dynamic neuronal feedatasetack. These methods are employed to precisely evaluate and improve the diagnostic dependability of AI. The first approach enables the neural network to generate a measure of its dependability

rather than a solitary prediction result. On the other hand, the latter approach refers to a computational neural feedforward cell, which allows the neural network to transmit information from the output layer to the shallow levels. This allows the neural network to selectively stimulate certain neurons. To evaluate the effectiveness of these two techniques, the authors provided training on 4,151 EKG signals that have three CHDs. The trained CHDNets were assessed using an internal test set including 1,037 EKG movies and an external set of 692 videos gathered from various cardiovascular imaging devices. Each EKG film is linked to a distinct patient. This study demonstrates the impact of BI on various neural network structures and quantifies the significant disparity in performance between internal and external test sets.

There are thirty-five different types of CHD that involve various abnormalities in the heart, such as defective, incomplete, or missing sections of the heart. Likewise, there are valves that allow leakage and openings in the heart chamber's partitions. *Luo et al. (2017)* proposed a novel DL model called the Cardiac Deep Learning Model (CDLM) that can efficiently and accurately identify this anomaly in its early stages using CT-scanned images. The authors employ a segmentation model to divide the four chambers of the heart, followed by the blood pool stage. To extract connection data and determine the categories of all the boats, the authors use a graph-matching technique. A publicly accessible dataset consisting of 68 CT images of the heart has been used for experimental work. According to the authors, the proposed strategy yielded superior outcomes in comparison to the previously published results.

A recent publication on CHD recognition using DL is proposed in *Garcia-Canadilla et al. (2020)*. This article presents an ML-based CHD recognition method called ML-CHDPM. This method investigated the integration of long-short-term memory (LSTM) and specific attention mechanisms (AM). This work also introduces a method where CNN, Bi-directional LSTM, and AM are combined. With this strategy, improvements in the results have been noticed. The method uses the ECG signals obtained from pregnant women. The ML-CHDPM is trained on a comprehensive dataset consisting of 33 applicants. The first 15 are those mothers who have an abnormal kid in the feuoutus and then the next 18 are normal. The first 15 patient group are again classified into CHD and severe CHD patients. The proposed model captures some good patterns and relationships in the data, which resulted in precise classification. The model has been evaluated with sensitivity, accuracy, specificity, and area under the receiver operating characteristic curve. According to the authors, previously reported results have been improved with the proposed model.

*Jia et al., (2024)* introduced an approach they called multilayer deep detection perception (DLDDP). The article's authors used ultrasonic images for their analysis. Both healthy and CHD patients were considered in the training and testing phases. The model extracts features through a multilayer deep learning framework with multiple perceptron layers. The authors claim excellent performance with the proposed model on the SOA dataset. The authors reported their results in root means square error (RMSE).

DL based methods are not new to ML-based applications, but we notice that for CHD recognition, these methods are new. The use of these methods has been sporadic so far. We still need to evaluate their complete potential for CHD recognition. Our research focusses

on the latest advancements in employing AI for CHD recognition over the past eight years. The following paragraphs present a compilation of observations that emerged from this study:

- **Datasets problems:** The lack of public datasets is a major barrier to exploring CHD classification using ML. Very few datasets are available for CHD recognition, with limited data. Most of these datasets are not good for DL based methods, as the data required for DL is too much. Table 1 provides details about the datasets reported so far. The reported datasets are for all kinds of data (X-rays, POX, EKG, ECG, *etc.*). Collecting these datasets within the first few days of birth is rare. Conversely, early recognition of CHD is crucial. One of the main reasons for the scarcity of datasets is the difficulty of acquiring them. The time available for researchers to obtain and collect data is very short. People normally hesitate to expose their kids to the equipment and rays. The other reasons are privacy and ethical concerns. It is not easy to make the data available for research without proper written consent from the parents and guardians. Table 1 shows that very few datasets are available for free download. Most DL algorithms require a significant amount of data for training and testing. While the number of CHD datasets has grown over time, only a small number of them are currently suitable for implementing a DL framework. Hence, it is essential to establish an initial measure by creating a comprehensive dataset that encompasses all the different aspects, such as session duration, ethnicity, age, gender, and many other complex parameters. However, we have noticed some gradual improvement in the quality of ground truth data over the last couple of years.

  In the seminal work on CHD recognition, errors were present in the ground truth data. The manual process is one possible cause of these errors. We contend that the creation of ground truth data may not be a forefront research field, but it remains just as crucial as any suggested solution for computer vision applications. Accurate verification and assessment of any algorithm is impossible without the prior preparation of ground truth data. Improved analysis can only be achieved with more accurate ground truth data. The data's accuracy and preparation are dependent on the specific task at hand. In the context of 3D image reconstruction, it is crucial to precisely identify the characteristics of the ground truth data for every task. Generating accurate data for certain tasks, such as gender, ethnicity, and expression classification, is quite straightforward. Humans or machines can either annotate or automate the labeling procedure. Generating accurate reference data for CHD might be challenging. Previously, a human performed manual annotation to generate ground truth. In this procedure, a person assigns a precise label. Generating accurate data with this approach is simple for smaller databases; however, as the size of the databases grows, it becomes a laborious and time-consuming process. Furthermore, these methods are more prone to human mistakes. There have been more than 35 types of CHD introduced so far. Classification and recognition of 35 CHDs is not easy for medical practitioners and neonatologists. This is also a possible reason why most of the ground truth data contains errors.
- **Limitations of existing feature extraction:** Limitations of existing feature extraction methods are also a problem computer vision and ML experts are facing when addressing

CHD with conventional ML. The activities of pre-processing and feature extraction are crucial in the development of an ML algorithm. The choice of the best appropriate technique for pre-processing is contingent upon the characteristics of the dataset. Out of various strategies, the one that is most appropriate for a particular acquisition typically fulfils the intended objective. The methods reported so far across various modules exhibit a wide range of variation. We have noticed comparable observations for several feature extraction strategies. The establishment and implementation of standardised techniques for reporting remain unresolved and unfinished. One possible reason is the diversity of data for CHD recognition. For example, a method that works for images will, of course, not be suitable for audio or tabular data.

- **The classification module presents challenges:** The detection and diagnosis of CHD have been a prominent focus of research for an extended period of time. Researchers have obtained highly satisfactory results despite using a limited amount of data for both training and testing. Researchers in this field are investigating numerous classifiers. The study found that backpropagation neural networks, SVM, and discriminant analysis (specifically linear) outperformed other methods. Afterwards, researchers employed Naive Bayes, random forest, K-nearest neighbour, and multilayer perceptron. However, the introduction of optimised deep neural networks has significantly enhanced the most advanced outcomes. Enhanced use of deep CNNs can optimise outcomes for large datasets.

- **Limitations of available systems:** We argue that a system's performance is highly dependent on the quality of its training data. In CHD recognition, it is the training data and certain extracted features, that significantly affect the performance of a system. High-quality data training enhances the performance of a system. However, to perform accurately, most existing systems must meet a specific set of requirements. If the system fails to meet some of these constraints, it may produce inaccurate results, which could ultimately lead to incorrect disease detection. For instance, overfitting is a common issue with most DL-based methods, especially conventional ML methods. Researchers need to consider designing adaptive systems with more flexible requirements. Additionally, adapting some generalised methods to work in heterogeneous environments is crucial. To improve efficiency, in-depth knowledge of the methods and proper use of the tools are also required.

- **Evaluation metrics:** We can use various metrics to evaluate and compare different models for CHD recognition. Four terminologies provide the basis for these measurements:

$$F_1 = 2 \cdot \frac{\text{Precision} \cdot \text{Recall}}{\text{Precision} + \text{Recall}}. \tag{1}$$

1. The number of correctly detected infected samples is known as true-positive (TP).
2. True-negative (TN) refers to correctly identified unhealthy data samples.
3. Similarly, we refer to false-positives (FPs) as instances where we mistakenly label healthy samples as infectious ones.
4. Lastly, an incorrect classification of infected samples as healthy is known as a false-negative (FN).

CHD accuracy is defined as the proportion of correct classifications ($TP + FP$) to the total number of classifications ($TP + FP + TN + FN$). Precision is a measure of how accurately infected samples are identified. We calculate it by dividing the number of correctly detected infected samples by the total number of samples identified as infected (the sum of TP and FP). In a similar way, recall is defined as the ratio of TP to the total number of infected samples, which is the sum of TP and FN as given in the Table 3. Finally, the F-measure is a statistical metric that calculates the harmonic mean of precision and recall as show in the Eq. (1). Based on these terms, some further methodologies are defined, such as accuracy ($CHD\_acc$), sensitivity, ROC, and specificity. Different terms are used, as can be seen from Table 2 by different authors.

- **Research methods progress and comparison:** CHD recognition and classification using ML and DL is an active area of research. However, we must comment here that it is not a mature research area specifically for ML experts. The number of articles reported on the topic is not excessive. Researchers have used very limited methods to explore this topic. Table 2 reports the performance results for every method in the last eight years. Table 2 presents a concise overview of the CHD recognition research conducted between 2016 and 2024. The data shown in Table 2 clearly demonstrates a gradual improvement. Upon examining the results in Table 2, it is evident that the performance for CHD recognition differs between classic ML approaches and recently developed DL methods. Based on the data from (*Cornforth & Jelinek, 2016*; *Chang et al., 2022*; *Pace et al., 2015*; *Chessa et al., 2022*; *Hoodbhoy et al., 2021*; *Morris & Lopez, 2021*), it is evident that DL modelling methods outperform standard ML-based methods.

  Furthermore, in certain instances, systems that rely on influence demonstrate superior performance compared to methods based on DL. Hence, we assert that there is a pressing need for a more comprehensive comprehension of the DL algorithm techniques and their applications. DL algorithms demonstrate significantly enhanced outcomes for difficult databases, such as (*Chen et al., 2024*; *Pachiyannan et al., 2024*). Nevertheless, DL has exhibited significantly superior performance when applied to the identical set of datasets. Table 2 demonstrates a varied response in CHD recognition when it comes to the performance of classic ML techniques. Hybrid models demonstrate significantly improved outcomes, as evidenced by the data presented in Table 2.

- **Knowledge transfer (KT) and data augmentation:** We anticipate that CHD recognition and classification are shifting towards novel DL methods in search of emerging trends in computer vision. DL approaches encounter training difficulties due to limited ground truth data. Accurate knowledge transfer (KT) is a potential solution to this issue (*Hoffman et al., 2014*). One recommendation is to explore options such as self-directed learning and supervised learning (*Zhou, 2018*). Another potential area for improvement includes the implementation of data augmentation techniques (*Wang et al., 2015*) and the utilisation of foveated architectural methodologies (*Karpathy et al., 2014*). In DL architectures, data augmentation helps to mitigate the issue of limited data.

  Furthermore, we would like to mention that heterogeneous domain adoption is a relatively underexplored area for knowledge transfer in DL. KT is highly effective in transferring knowledge from the training phase to the testing phase, especially when

**Table 3  Confusion matrix.**

|  | Predicted | |
| --- | --- | --- |
|  | **Positive** | **Negative** |
| **Actual positive** | **TP** | FN |
| **Actual negative** | FP | TN |

qualities exhibit some degree of variation. This significantly reduces the amount of work required to label the training data. Recent advancements in deep learning techniques indicate a need for further exploration of concepts such as temporal pooling, LSTM networks, optical flow frames, and 3D convolution in the analysis of CHD data. Although academics are already investigating some of the strategies indicated above, further study is necessary to enhance the performance of these activities for CHD recognition.

## Healthcare challenges in MLICs

Healthcare Challenges in middle- and low-income countries MLICs often face significant barriers to effective healthcare delivery, including limited financial resources, inadequate infrastructure, and a shortage of skilled healthcare professionals. These factors contribute to delayed diagnosis and treatment, especially for chronic conditions like congenital heart disease (CHD). Machine learning has the potential to bridge some of these gaps by providing scalable, cost-effective diagnostic tools.

1. **Challenges specific to MLICs:**

   - Data availability and quality: In many MLICs, healthcare data is scarce, poorly documented, or fragmented across different platforms. The lack of standardized, high-quality medical datasets makes it challenging to develop and train accurate ML models.
   - Technological infrastructure: Many healthcare facilities lack the technological infrastructure needed to deploy advanced ML-based systems, including reliable internet access, modern diagnostic equipment, and sufficient computational resources.
   - Limited ML expertise: A shortage of trained data scientists and ML practitioners in MLICs makes it difficult to locally develop and implement machine learning solutions. Reliance on external expertise can slow the adoption of ML in healthcare.
   - Healthcare disparities: ML systems trained on data from high-income countries may not be generalizable to MLICs, where the distribution of diseases, socioeconomic factors, and healthcare practices are different. This can lead to biased or inaccurate predictions in these regions.

2. **Applications of ML in MLICs healthcare systems:** Despite the challenges, ML has demonstrated its potential to transform healthcare in resource-constrained settings. Several successful ML applications can serve as a foundation for CHD diagnosis in MLICs:

- ML-based diagnostic tools can be integrated into telemedicine platforms to provide real-time, remote consultations. This is especially valuable in rural or underserved areas where access to specialized cardiologists is limited.
- Automated image analysis: ML algorithms, such as convolutional neural networks (CNNs), can analyze medical images (*e.g.*, echocardiograms or X-rays) to detect heart abnormalities. In regions where radiologists or cardiologists are scarce, these tools can assist non-specialist healthcare workers in making accurate diagnoses.
- Predictive analytics: ML models can analyze patient records and risk factors to predict the likelihood of CHD, enabling earlier interventions. These tools can also help prioritize high-risk patients for treatment in overburdened healthcare systems.
- Mobile health (mHealth) applications: Mobile phone-based ML applications can be used to collect and analyze patient data remotely, facilitating early diagnosis and monitoring of CHD patients without the need for frequent hospital visits. Mobile health solutions are particularly effective in MLICs, where mobile phone penetration is high, even in remote areas.

3. **Ethical implications and risks:** While ML can improve healthcare access and outcomes in MLICs, there are several ethical concerns that must be addressed:

- **Data privacy:** Many MLICs lack robust data protection laws, and the collection and use of sensitive health data by ML systems could raise privacy concerns. Ensuring patient consent and protecting data from misuse is critical.
- **Algorithmic bias:** If ML models are trained on data from high-income countries or specific ethnic groups, they may not perform well in MLIC populations, potentially leading to misdiagnoses or unequal access to care.
- **Human oversight:** While ML can assist in decision-making, it is important to ensure that healthcare professionals remain involved in the diagnostic process to avoid over-reliance on algorithms, especially in critical areas like CHD diagnosis.

## CONCLUSION

The integration of AI into the field of CHD research and treatment has demonstrated substantial progress and potential over the last decade. This survey has highlighted the various AI techniques applied in CHD, the significant advancements made, and the impact these technologies have had on improving the diagnosis, prognosis, and treatment of CHD.

### Summary of key findings

AI, particularly ML and DL, has shown remarkable effectiveness in analyzing complex medical data. We believe that the next revolution in the medical sector is the incorporation of AI in healthcare. In addition to easing the workload of physicians, AI provides new directions for researchers in the fields of AI and ML. Pediatric cardiology, with its demand for high cognitive ability and interpretive skills, is particularly well-suited for AI integration. AI has been effectively incorporated into various aspects of pediatric cardiology, including clinical examination, image analysis, diagnosis, prognosis, risk evaluation, precision medicine, and treatment. The emergence of AI has enhanced accuracy and precision in

the medical field. However, it is important to note that AI in medicine is still evolving and faces various challenges and constraints. Despite these obstacles, we confidently believe that AI, at its current rate of progress, will continue to improve and simplify methods in pediatric cardiology.

This article provides a comprehensive analysis of methods used for identifying CHD, along with a compilation of publicly accessible datasets. We investigated different aspects of existing solutions for CHD recognition with ML. First, we reviewed state-of-the-art (SOA) methods based on simple and hand-crafted representations. We then moved on to recently introduced DL frameworks. Additionally, we conducted a thorough evaluation of the performance of the SOA methods reported so far. Ultimately, we have reported numerous unresolved issues in the recognition of CHD. Specifically, we anticipate more assessments of the recently introduced DL algorithms on the most challenging datasets. We hope that this overview study will serve as a catalyst for the exploration of novel research avenues in this field and advance methodologies by providing a comprehensive list of datasets, methods, and algorithms.

## Final thoughts on the impact of AI in combating CHD

The future of AI in CHD research and treatment is promising. Emerging technologies such as federated learning and edge AI offer new possibilities for handling sensitive medical data and enabling real-time analysis. Integrating AI with other advanced technologies, such as robotics and the Internet of Things (IoT), can further enhance CHD care by providing precise surgical assistance and continuous health monitoring.

Multidisciplinary collaborations will be key to driving innovation in this field. Bringing together AI researchers, clinicians, bioinformaticians, and industry partners will foster the development of clinically relevant AI solutions that address real-world challenges. These collaborations will help bridge the gap between technological advancements and practical applications, ensuring that AI tools are designed with patient outcomes in mind.

Ultimately, AI holds the potential to revolutionize CHD care by moving towards personalized medicine. Using patient-specific data, AI can tailor treatment plans to individual needs, improving efficacy and outcomes. Predictive models can identify which patients are most likely to benefit from specific treatments, reducing trial-and-error approaches, and enhancing the precision of medical interventions.

In conclusion, while significant progress has been made, the journey toward fully integrating AI into CHD care is ongoing. Continued research, innovation, and collaboration are essential to unlock the full potential of AI in combating congenital heart disease. By addressing current challenges and embracing future opportunities, AI can transform the care of CHD, improving the lives of countless patients around the world.

### Funding

This research was supported by Nazarbayev University, Kazakhstan, through Social Policy Research Grant Program (SPRG) grant. The funders had no role in study design, data collection and analysis, decision to publish, or preparation of the manuscript.

### Grant Disclosures

The following grant information was disclosed by the authors:
Nazarbayev University, Kazakhstan, through Social Policy Research Grant Program (SPRG).

### Competing Interests

The authors declare there are no competing interests.

### Author Contributions

- Khalil Khan conceived and designed the experiments, performed the experiments, analyzed the data, performed the computation work, prepared figures and/or tables, authored or reviewed drafts of the article, and approved the final draft.
- Farhan Ullah conceived and designed the experiments, performed the experiments, analyzed the data, performed the computation work, prepared figures and/or tables, authored or reviewed drafts of the article, and approved the final draft.
- Ikram Syed conceived and designed the experiments, performed the experiments, analyzed the data, performed the computation work, prepared figures and/or tables, authored or reviewed drafts of the article, and approved the final draft.
- Hashim Ali conceived and designed the experiments, performed the experiments, analyzed the data, performed the computation work, prepared figures and/or tables, authored or reviewed drafts of the article, and approved the final draft.

### Data Availability

   This is a literature review.

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
