# Peer review of "Accurately assessing congenital heart disease using artificial intelligence"

_PeerJ Computer Science, doi:10.7717/peerj-cs.2535_

## Round 0.1 · original submission · Major Revisions

Two reviewers have suggested that the paper is written well and that the topic is interesting and within the scope of the journal. The reviewers have also highlighted the need to improve the overall English composition. Furthermore, the research questions need to be clearly identified. Similarly, a comparison must be made with recent surveys on the topics.

I am giving a major review on this and providing an opportunity for the authors to improve the paper and carefully address the comments and concerns of the reviewers.

Reviewer 1 ·

Basic reporting

The language use is good and grammar is correct. However, new references must be added. The structure and organization of the paper is also acceptable, and little efforts are needed to improve contents structure and enhance readability.

Experimental design

The article is with in the scope of the journal. The contributions are marginal (limited to few ML related papers for CHD and their summaries) and has not merit for publication. The ML and DL methods are also not described well.

Validity of the findings

Albeit, there are few rela datasets that are explained in the paper, However, the authors does not have any proposed method to use these datasets. There are no findings based on real datasets; and the conclusions are not well stated.

Additional comments

I have the following comments:

1) The abstract is not coherent. It would be good if authors can write a sentence describing numerical results and improvement over other methods.

2) The introduction, e.g., should lead the way throughout the paper. In addition, the benefits coming from this research should be made clearer in the introduction and throughout the paper. I believe, this section needs significant efforts to make things and contribution clearer and flow of the contents.

Furthermore, briefly describe the major contributions in bullet form, just before the organization paragraph.
Add the final paragraph that describes the entire organization of the paper.
This section is a bit long. I recommend removing the related work and put them either in the background section or in a separate section, i.e. related work.

3) I suggest summarising the related works into a table with respect to their characteristics. Moreover, authors should put their proposal into this table for easy comparison.

This will make it clearer to readers, and they will be able to see what was missing in the literature; and how this is addressed in this paper.

4) Although, few real datasets are noted in the paper and the authors have discussed them; however, I cannot see any contributions of the authors. It should be better to propose a model or ML technique that uses these datasets.

5) Some sections has repeated text or the discussion is too-detailed; and should be reduced. Furthermore, some figures need appropriate references.

6) The conclusion section also needs significant revisions. It should briefly describe the findings of the study and some more directions for further research.

7) Proofread the article to ensure appropriate use of English grammar, tenses, and punctuations. Longer sentences should be broken out into smaller ones. There are also some linguistic issues that should be corrected. The use of article "the" is redundant and somewhere missing.

8) The author have discussed ML and DL for CHD, but there is no discussion of various related works in terms of results? There are few metrics such F1, TF, FP etc. are discussed but never used in the study.

9) Regarding Table 2, I suggest authors to study their results and provide a comparative analysis in their study. Unfortunately, summarizing few papers does not make any publishable contribution.

Cite this review as

Reviewer 2 ·

Basic reporting

Overall the manuscript is very interesting and informative. Overall it is well written, however, I would a few minor changes before accepting it. Firstly, Section 2 should be restructured by putting 2. Survey Methodology inside the introduction as one of the subsections. The contribution along with a comparison with the existing surveys should be provided in the introduction. Section 3 also needs restructuring and must be organized in a better way by separating the datasets and challenges.

Experimental design

More details on the rejection criteria would help the readers.

Validity of the findings

Section 5.2 Future Work must be extended by adding more details on the areas identified. The future directions identified in the paper are very generic and adding more details would help the readers

Cite this review as

·

Basic reporting

English writing is good and structure can be improved. Furthermore, latest references and related work is missing.

Experimental design

The methodology of the research is clearly presented and falls with in the PeerJ Computer Science Journal scope. Methods are NOT described with clear description and must be improved.

Validity of the findings

The findings of the study lacks detailed description like what research questions have been answered.

Additional comments

The following suggestion must be followed to improve the paper.

1. Improve the abstract while describing clearly the problem and methodology.

2. I suggest adding some research questions and then discuss in the findings that how the authors answered to the questions. This will increase readability of the research to a broader audience.

3. Adding transitional phrases between the introduction and other sections will enhance the paper's coherence, providing a clearer guide through the complex subject.

4. The literature can be improved further by discussing state-of-the-art works. There is rich literature on the topic that should be investigated and compared.

5. The reference section should be formatted uniformly according to a specific citation style, which will improve the paper's academic rigour and professionalism.

6. Proofreading is important, and I suggest proofreading from a native English speaker.

---

## Round 0.2 · Minor Revisions

The two reviewers agree that the paper highlights signficant potential of AI in heart diseases. One reviewer (R4) has provided some minor comments that are important to be addressed. I would be glad to accept the paper should the authors be willing to address the comments of the reviewer.

Reviewer 2 ·

Basic reporting

My comments are fully addressed and I am satisfied with the changes.

Experimental design

My comments are addressed

Validity of the findings

My comments are addressed

Additional comments

My comments are addressed

Cite this review as

·

Basic reporting

The authors have not addressed my concerns and the paper lacks to discuss important concepts. Furthermore, the quality of the paper does not meet the standard for PeerJ journal.

Experimental design

The aim and scope of the study are not discussed well. The evaluation is weak, and the proposed method is not described well.

Validity of the findings

The experiments are weak and does not fully justify the findings of the study.

Reviewer 4 ·

Basic reporting

The paper is generally well-written. The title appears to be a placeholder text and not reflective of the paper's content, which needs to be corrected. The abstract needs to be revised and enhanced by providing a clearer explanation of the research contributions and explicitly outlining the study's goals. The introduction sets a solid foundation, emphasizing the importance of congenital heart disease (CHD) and the role of machine learning (ML) in its diagnosis and treatment. The literature review is thorough, and the discussion on existing ML algorithms and datasets is well-documented. However, some sections could benefit from additional citations to support claims, especially in regard to healthcare disparities and the impact of ML in low-resource settings. The paper seems a bit lengthy, I believe.

Experimental design

As this is a review paper, the structure follows the expected format for such works, summarizing and synthesizing recent advances in the field. The selection of the past eight years as the timeframe is appropriate and justified. The paper could benefit from a more explicit description of how studies were chosen for inclusion in the review, detailing any inclusion/exclusion criteria and addressing potential biases in the selection of datasets and algorithms discussed. This would improve the transparency and reproducibility of the review.

Validity of the findings

The comparison of strengths and weaknesses of ML algorithms is ..., though it could benefit from a more detailed quantitative analysis of the performance metrics discussed. Additionally, while the future directions for ML in CHD diagnosis are promising, the paper could include more concrete recommendations or challenges to be addressed in practical implementation.

Additional comments

The title needs to be updated to accurately reflect the subject matter of the paper.
Consider providing a flowchart or diagram summarizing the key ML methods and parameters discussed for clarity.
While the abstract mentions middle- and low-income countries, there is little detailed discussion about specific ML applications or challenges in these regions. Expanding on this would strengthen the paper.
Include a brief section on the ethical implications of using ML in CHD diagnosis, such as data privacy, algorithmic bias, and the need for human oversight.

Cite this review as

---

## Round 0.3 · accepted · Accept

The authors have fairly responded to the minor comments.